# RETHINKING TABLE INSTRUCTION TUNING

## ABSTRACT

Recent advances in table understanding have focused on instruction-tuning large language models (LLMs) for table-related tasks. However, existing research has overlooked the impact of hyperparameter choices and lacks a comprehensive evaluation of the out-of-domain table understanding ability and the general capabilities of these table LLMs. In this paper, we evaluate these abilities in existing table LLMs, and reveal significant declines in both out-of-domain table understanding and general capabilities compared to their base models. Through systematic analysis, we show that hyperparameters, such as learning rate, can significantly influence both table-specific and general capabilities. Contrary to the existing table instruction-tuning works, we demonstrate that smaller learning rates and fewer training instances can enhance table understanding while preserving general capabilities. Based on our findings, we introduce *TAMA*, a **TA**ble LLM instruction-tuned from LLa**MA** 3.1 8B Instruct, which achieves performance on par with, or surpassing GPT-3.5 and GPT-4 on table tasks, while maintaining strong out-of-domain generalization and general capabilities. Our findings highlight the potential for reduced data annotation costs and more efficient model development through careful hyperparameter selection.

## 1 INTRODUCTION

Recent years have witnessed a paradigm shift to data-driven methods for table understanding. Researchers have instruction-tuned various LLMs, particularly the open-source models from LLaMA family (Touvron et al., 2023; Dubey et al., 2024) to improve their ability on handling table-related tasks, such as table question answering (Table QA) (Nan et al., 2022), and table fact verification (Chen et al., 2019), and pushing the state-of-the-art performance on various table benchmarks (Zhang et al., 2024a;b).

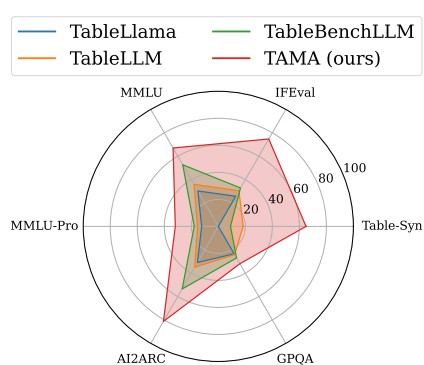

Figure 1: Performance comparison between our proposed model *TAMA* and the existing table LLMs on out-of-domain table understanding and general benchmarks.

However, existing research has been influenced by the lack of transparency in closed-source LLMs, which often claim to be trained on large-scale datasets without revealing the detailed training process. As a result, open-source efforts have tended to follow these closed-source models by focusing primarily on large-scale datasets (Zhang et al., 2024a), while overlooking the crucial influence of hyperparameter choices. In addition, existing works lack the discussion of how these table LLMs perform on out-of-domain table understanding tasks, and how they compromise their general ability when specializing on table tasks. We argue that out-of-domain table understanding is crucial for table LLMs, as it reflects how well these models generalize to unseen table tasks. In addition, the general capabilities of these models are still important for handling table-related tasks. For instance, instruction following is crucial in real-world applications where end-users may request specific input-output formats (e.g., The user may request the model to return the answer in JSON). Additionally, stronger reasoning capabilities and comprehensive general knowledge can enhance these models' ability to handle diverse

Table 1: Information of existing table instruction tuned models. For "Data Source", "S" and "R" represent synthesized data and real data, respectively. †: a variant based on the LLaMA 2 7B model.

| Model | Base Model | Learning Rate | Epochs | Data Size | Data Source | Open-Source? |
|---|---|---|---|---|---|---|
| TableGPT (Zha et al., 2023) | - | - | - | - | - | ✗ |
| Table-GPT (Li et al., 2023) | GPT-3.5 | - | - | 13K | S | ✗ |
| TableLLaMA (Zhang et al., 2024a) | LongLoRA 7B† | 2e-5 | 6 | 2M | R | ✓ |
| TableLLM (Zhang et al., 2024b) | CodeLLaMA 7B & 13B Instruct | 2e-5 | 6 | 309K | R + S | ✓ |
| TableBenchLLM (Wu et al., 2024) | LLaMA 3.1-8B & others | 2e-5 | 3 | 20K | S | ✓ |

scenarios, such as interpreting user queries and reasoning over complex data. Therefore, having an understanding of these table LLMs' general capabilities gives us a comprehensive understanding of these models' limitations in our practical usage.

In this paper, we first evaluate the existing table LLMs in terms of their out-of-domain table understanding ability and their general abilities. We reveal that existing table LLMs suffer from a significant decline in terms of these abilities compared to their base models. Sometimes, the performance decline on general reasoning benchmarks, such as AI2ARC, can be up to 20 percentage.

We then select the latest LLaMA 3.1 8B Instruct model, and proceed to explore how hyperparameter choices influence the model's performance. Our analysis reveals that learning rate plays a crucial role in shaping the model's table understanding ability and influencing the model's general ability. A large learning rate, as seen in the existing table LLMs, compromises the model's general capabilities and leads to suboptimal table understanding performance. On the other hand, a small learning rate, while effectively preserving the model's general capabilities, fails to sufficiently improve its table understanding ability. In addition, we find that it is possible to achieve strong table understanding ability with a much smaller amount of training data – for instance, 2,600 in Section 4. Our training size is significantly smaller compared to the two million instances used by TableLLaMA (Zhang et al., 2024a), and ten times smaller than that of TableBenchLLM (Wu et al., 2024), highlighting the potential to reduce annotation costs in future model development. We also explore the effects of epoch numbers and the task synergy, and discuss our findings in Section 3.

Based on our findings, we carefully select the hyperparameters and instruction-tune the LLaMA 3.1 8B Instruct model, resulting in *TAMA*, which demonstrates strong table understanding ability and general capabilities (Figure 1).

In summary, our contributions are three folds:

- We examine the existing table LLMs and reveal that these table LLMs do not generalize to out-of-domain table tasks and show compromised general capabilities compared to their base model.

- We reveal the impacts of the often-ignored hyperparameter selection such as the learning rate, number of training instances, etc. We find that the commonly-adopted learning rate can be too large, and may lead to suboptimal table understanding performance and compromises the model's general capabilities. In addition, we can achieve strong table understanding ability with a much smaller amount of training data compared to the existing works.

- Based on our findings, with careful hyperparameter selection, we instruction-tune LLaMA 3.1 8B Instruct model with 2,600 table instruction data. As an 8B size model, our resulting model, *TAMA* achieves performance on par with, or even exceeding GPT-3.5 in table understanding tasks, and in some cases surpasses GPT-4, while retaining the general capabilities of its base model. Moreover, *TAMA* exhibits strong out-of-domain table understanding and general capabilities (Figure 1).

In the following sections, Section 2 evaluates the existing table LLMs in terms of their out-of-domain table understanding ability and general capabilities. Section 3 explores how the hyperparameter choices shape the model's ability. Based on our findings in Section 3, we build our model, *TAMA* in Section 4.

Table 2: Details of the benchmarks upon which we evaluate the existing table LLMs. We report the performance on the main set for GPQA and the challenge set for AI2ARC.

| Evaluation Datasets | Category | # Shots | Task Type | Metrics |
|---|---|---|---|---|
| Table-Syn[2] (Li et al., 2023) | Table understanding | - | Generation | Acc |
| IFEval (Zhou et al., 2023) | Instruction Following | - | Generation | Instance-level Acc |
| MMLU (Hendrycks et al., 2021) | General | 5-shot | Multi-Choice | Acc |
| MMLU $_{Pro}$(Wang et al., 2024) | General | 5-shot | Multi-Choice | Acc |
| AI2ARC (Clark et al., 2018) | Reasoning | 0-shot | Multi-Choice | Acc |
| GPQA (Rein et al., 2023) | Reasoning | 0-shot | Multi-Choice | Acc |

Table 3: Performance comparison between the existing table LLMs (second row) and their base models (first row). †: A variant of LLaMA 2 7B model.

| | Table-Syn | IFEval | MMLU | MMLU$_{Pro}$ | AI2ARC | GPQA |
|---|---|---|---|---|---|---|
| LongLoRA 7B$^{†}$ | 2.40 | 31.41 | 44.22 | 17.51 | 42.24 | 23.66 |
| TableLLaMA | 0.00 | 25.78 | 30.27 | 12.33 | 30.89 | 23.44 |
| Δ | ↓ 2.40 | ↓ 5.63 | ↓ 13.95 | ↓ 5.18 | ↓ 11.35 | ↓ 0.22 |
| CodeLLaMA 13B Instruct | 33.40 | 48.32 | 44.69 | 19.66 | 48.72 | 24.78 |
| TableLLM | 18.40 | 30.46 | 35.90 | 15.36 | 34.81 | 24.11 |
| Δ | ↓ 15.00 | ↓ 17.86 | ↓ 8.79 | ↓ 4.30 | ↓ 13.91 | ↓ 0.67 |
| LLaMA 3.1-8B | 13.40 | 32.13 | 62.08 | 13.86 | 74.40 | 28.12 |
| TableBenchLLM | 9.00 | 32.85 | 52.67 | 17.84 | 53.50 | 27.01 |
| Δ | ↓ 4.40 | ↑ 0.72 | ↓ 9.41 | ↑ 3.98 | ↓ 20.90 | ↓ 1.11 |

## 2 EVALUATION OF EXISTING TABLE LLMS

### 2.1 EXPERIMENTAL SETUP

**Models to Evaluate.** Table 1 provides a comprehensive overview of the existing table LLMs. As we do not have access to the closed-source table LLMs, we focus on the evaluation of the open-source ones, including TableLLaMA (Zhang et al., 2024a), TableLLM (Zhang et al., 2024b), and TableBenchLLM (Wu et al., 2024). All of these open models are fine-tuned with all parameters being updated.

**Evaluation Datasets.** Table 2 provides the datasets on which we test these table LLMs in terms of their out-of-domain table understanding ability and their general capabilities. We choose Table-Syn (Li et al., 2023) to test these table LLMs' out-of-domain table understanding ability, as none of them has been fine-tuned on this dataset.

### 2.2 FINDINGS

*Existing Table LLMs possess limited out-of-domain table understanding ability.* In Table 3, all the existing table LLMs suffer from performance drops on Table-Syn compared to their base models. Though these table LLMs achieve SOTA performance on various benchmarks (Zhang et al., 2024a;b), such a performance decline reveals their limited out-of-domain table understanding capabilities, which aligns with the findings by Zheng et al. (2024).

*Existing Table LLMs demonstrate poor instruction-following ability.* In Table 3, both TableLLaMA and TableLLM show significant drops in performance on IFEval (Zhou et al., 2023), with accuracy declines of 5.63 and 17.86, resulting in a score of 25.78 and 30.46, respectively. While TableBench-LLM maintains a similar score to its base model (32.85 compared to 32.13 for LLaMA 3.1-8B), this performance is still limited compared to 83.57 by GPT-4 reported by Zhou et al. (2023). At such low instruction following scores, existing table LLMs cannot consistently follow instructions such as "return the answer in JSON format" as shown in Table 6 in Section 4.3 and Tables 17 to 19 in

Appendix E, limiting the model's usage if the end users need data extraction that requires certain answer format.

*Existing table instruction tuning compromises models' general capabilities.* Existing table instruction-tuning methods lead to significant drops in accuracy on general benchmarks such as MMLU, AI2ARC, GPQA as shown in Table 3. For instance, compared to their base models, TableL-LaMA experiences a decline of 13.95 accuracy score on MMLU, while TableLLM and TableBench-LLM lose 8.79 and 9.41, respectively. Appendix B provides further discussion of the model's performance corresponding to each category in MMLU benchmark. On the general reasoning benchmarks such as AI2ARC, the drop can be as large as 20.90 for TableBenchLLM, showing that the existing table instruction tuning hurts their base model's reasoning ability. This limits the existing table LLMs' usage if there are general knowledge or reasoning involved in end users' request.

## 3 HYPERPARAMETER EXPLORATION

Table 1 reports the hyperparameters used in the existing table instruction tuning works. Although hyperparameter selection is often treated as technical detail and receives little attention, we demonstrate that these choices are crucial. The impact of factors such as learning rate, and number of epochs should not be underestimated, as they significantly influence both the table understanding and general ability. In the following subsections, Section 3.1 introduces the model and datasets used in our analysis experiments, Section 3.2 provides the findings and the choices we make that lead to our model in Section 4.

### 3.1 EXPERIMENTAL SETUP

**Models.** We conduct table instruction tuning with full parameter tuning using the 8B version of the LLaMA 3.1 Instruct model (Dubey et al., 2024) because of its superior general capabilities, especially its strong instruction following ability. Appendix C.1 provides detailed reasons for our model choice.

**Datasets.** We draw training data from three representative table understanding datasets in this section, **FeTaQA** (Nan et al., 2022), a free-form table question answering (Table QA) dataset; **HiTab** (Cheng et al., 2022), a short-answer Table QA dataset; **TabFact** (Chen et al., 2019), a table fact verification dataset. In Figure 2, we also report the model's performance on FEVEROUS (Aly et al., 2021), another table fact checking dataset, and on two general benchmarks, MMLU and IFEval introduced in Table 2.

### 3.2 ANALYSIS

**Learning Rate.** In Figure 2, we fine-tune the LLaMA 3.1 8B Instruct model using instruction data from TabFact, HiTab, and FeTaQA.

We find that *the learning rate plays a crucial role in determining model performance, as well as how well the model preserves its general capabilities.* In general, LLaMA 3.1 8B Instruct achieves the best performance when the learning rate is around 1.0e-6 and 5.0e-7. For instance, on TabFact, LLaMA 3.1 8B Instruct achieves its best performance (73.10) at a learning rate of 1.0e-6 with 1500 examples. Moreover, there is little to no decline in LLaMA 3.1 8B Instruct's performance on MMLU and IFEval with such learning rates. With a smaller learning rate such as 1.0e-7, though the model's performance on MMLU and IFEval can be well-preserved, the model's performance on table tasks such as FEVEROUS is suboptimal under the same setup (66.86 compared to 74.63 at a learning rate of 5.0e-6). In contrast, when the learning rate is too large, such as 1.0e-5, we observe a significant decline in the model's performance on both MMLU and IFEval, suggesting that a larger learning rate may hurt the model's general capabilities. We note that all the existing table LLMs use a large learning rate of 2e-5 (Table 1), which explains their compromised out-of-domain table understanding ability and general capabilities compared to their base models in Table 3.

**Number of Examples.** As the number of training instances increases, we find that *there is a period of quick learning followed by a period of marginal performance improvement.*

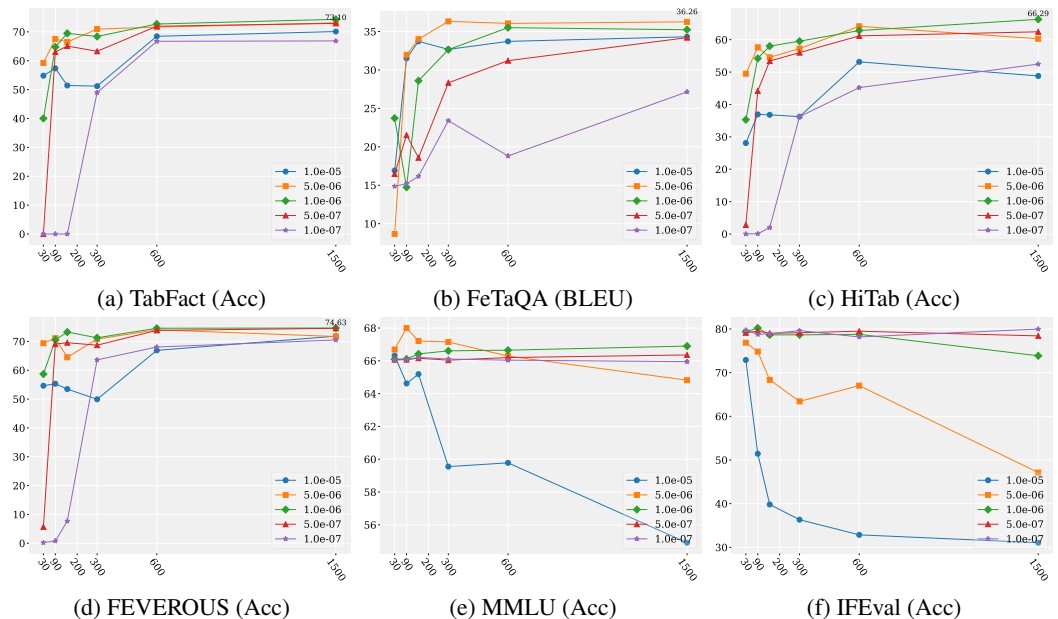

(a) TabFact (Acc)   (b) FeTaQA (BLEU)   (c) HiTab (Acc)

(d) FEVEROUS (Acc)   (e) MMLU (Acc)   (f) IFEval (Acc)

Figure 2: LLaMA 3.1 8B Instruct's performance (y-axis) with respect to the number of training instances (x-axis). We fine-tune the model for three epochs. We note that the learning rate plays a crucial role in shaping the model's capabilities, and the performance improvement beyond 200 examples seems marginal.

We observe in Figure 2 that on table tasks such as FeTaQA and HiTab, there is a period where the model's performance boosts up quickly, typically happening when tuning on the first 200 examples. Later, the performance improvement seems marginal. This aligns with the findings from Zhou et al. (2024) that the foundational LLM's performance can be improved with a limited amount of high-quality data in the instruction tuning stage. We hypothesize that with the first few hundred examples, the model is able to enhance its table reasoning ability quickly. After this point, the model's performance increase may primarily come from fitting the nuanced patterns in these datasets. Therefore, unlike the existing table LLMs which may involve up to two million training instances as seen in Table 1, we choose to train on 200 instances for each dataset in Section 4.

In addition, *we can achieve competitive or even SOTA performance with limited data*. On HiTab, with a learning rate of 1.0e-6 and 1,500 examples, we achieve an accuracy score of 66.29, outperforming the previous SOTA performance of 64.71 by TableLLaMA. On FEVEROUS, with 1,500 examples, we achieve a better score of 74.63 compared to 73.77 by TableLLaMA. Though the credit also comes from the LLaMA 3.1 Instruct model, which is much stronger compared to the LLaMA 2 model that TableLLaMA is tuned from, we highlight that TableLLaMA has used two million data points in its table instruction tuning stage, including the entire training set of TabFact, FeTaQA, and HiTab, while here we use around 7% of the entire training data for HiTab. Our analysis demonstrates that with a strong foundational model and a good choice of learning rate, we can achieve competitive performance on table understanding tasks with limited training instances.

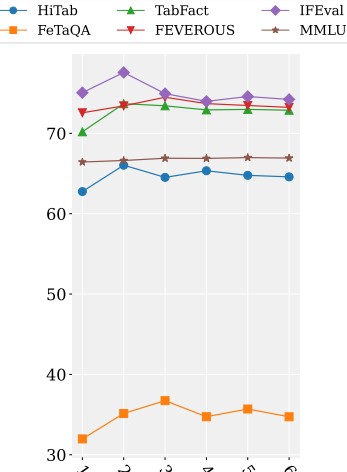

Figure 3: LLaMA 3.1 8B Instruct model's performance (y-axis) across different numbers of epochs (x-axis). We fine-tune the model on the 1,500 instruction pairs, with 500 pairs each from FeTaQA, HiTab, and TabFact, at a learning rate of 1.0e-6.

**Effects of Epochs.** Figure 3 illustrates the relationship between the performance of LLaMA 3.1 8B Instruct model and the number of epochs when we fine-tune the model on the 1,500 instruction pairs at a learning rate of 1.0e-6. The model demonstrates a decent performance on these table tasks within just one or two epochs. In the meantime, the model mostly preserves its performance on MMLU and IFEval, indicating that its general capabilities are not compromised too much while acquiring table reasoning ability. Beyond this point, there is no significant performance improvement, suggesting that extending training for more epochs yields diminishing returns or may even lead to overfitting. Therefore, we choose to train our model for two epochs in Section 4 instead of the commonly adopted six epochs by existing table LLMs as seen in Table 1.

**Effects of Multi-Task.** In Figure 4, we present the heatmap of model performance when fine-tuning the LLaMA 3.1 8B Instruct model on a single dataset (one of the datasets among FeTaQA, HiTab, and TabFact). We fine-tune the model for two epochs at a learning rate of 1.0e-6 with 500 instruction pairs, and then test it against the six datasets. Additionally, Figures 7 and 8 in Appendix C.4 present heatmaps across varying learning rates (from 1.0e-7 to 1.0e-5) and number of epochs (from one to six).

*There are synergy effects on these tasks.* The model achieves better performance when trained on the instruction pairs combined from all three datasets, compared to being trained on each of them separately. For instance, the accuracy on HiTab increases to 66.29, compared to 64.84 when trained only on HiTab as shown in Figure 7.

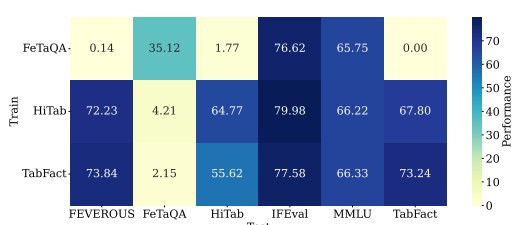

*There are inter-connections between different tasks.* In Figure 4, we note that fine-tuning solely on HiTab leads to a performance of 67.80 on TabFact, and fine-tuning solely on TabFact leads to a performance of 55.62 on HiTab, demonstrating a transfer of learned capabilities between these two tasks. However, this relationship is not universal as training on HiTab yields poor performance on FeTaQA, indicating that the overlap between certain tasks may be limited.

Figure 4: Heatmap when we fine-tune the LLaMA 3.1 8B Instruct model on a single dataset (y-axis) and test against the others (x-axis). In this plot, we fine-tune the model for two epochs at a learning rate of 1.0e-6 with 500 instruction pairs.

Based on these observations, we choose to fine-tune our model on a diverse range of tasks and datasets in Section 4. We provide further analysis across LLMs in Appendix C.2, and analysis in terms of LoRA and QLoRA in Appendix C.3. We provide further analysis regarding how the data features affect the model's performance degradation on general benchmarks in Appendix C.5.

## 4 *TAMA*

Based on our findings from Section 3, we start building our general table understanding model, *TAMA* by instruction tuning the LLaMA 3.1 8B Instruct model.

### 4.1 EXPERIMENTAL SETUP

**Hyperparameter Selection.** In Section 3, we find that with 200 instruction pairs, the model has already achieved competitive table understanding ability, and the performance gain after such a point is marginal. Moreover, tuning the model at a learning rate of 1.0e-6 for two epochs would enhance the model's table understanding ability while still maintaining its general ability. Therefore, we select 200 instruction pairs in the training set from each of the datasets in Table 4, and train the model at the learning rate of 1.0e-6 for two epochs.

**Dataset Splits.** As we use FeTaQA, HiTab, TabFact, FEVEROUS, MMLU, and IFEval in Section 3 for hyperparameter selection, we report their scores under the "Dev" category. In the test time, we test our model on the additional nine table understanding datasets in Table 4. Moreover,

Table 4: Datasets where we sample the instruction pairs to fine-tune the LLaMA 3.1 8B Instruct model. We randomly select 200 data points from each of these datasets in our table instruction tuning stage. We denote these datasets by their shorthands in Table 5.

| Task Category | Task Name | Dataset | Shorthand | #Size (Table/Sample) | Data Split | Metrics |
|---|---|---|---|---|---|---|
| Question Answering | Table QA | WikiTQ (2015) | W-T | 0.4K/4K | Test | Acc |
| | Table QA | WikiSQL (2017) | W-S | 5K/16K | Test | Acc |
| | Hybrid Table QA | HybridQA (2020) | Hyb | 3K/3K | Test | Acc |
| | Table QA | TATQA (2021) | TAT | 0.2K/0.7K | Test | Acc |
| | Highlighted Cells QA | FeTaQA (2022) | FeT | 2K/2K | Dev | BLEU |
| | Hierarchical Table QA | HiTab (2022) | HiT | 1K/1K | Dev | Acc |
| | Hierarchical Table QA | AIT-QA (2022) | AIT | 0.1K/0.3K | Test | Acc |
| | Table QA | TABMWP (2023) | TAB | 7K/7K | Test | Acc |
| Table Fact Verification | Fact Verification | TabFact (2019) | TaF | 2K/12K | Dev | Acc |
| | | InfoTabs[2] (2020) | Inf | 0.06K/0.5K | Test | Acc |
| | | FEVEROUS (2021) | FEV | 4K/7K | Dev | Acc |
| Dialogue Generation | Table Grounded Dialogue Generation | KVRET (2017) | KVR | 0.3K/0.8K | Test | Micro F1 |
| Data-to-Text | Highlighted Cells Description | ToTTo (2020) | ToT | 7K/8K | Test | BLEU |

Table 5: Evaluation results on the datasets listed in Table 4. "Base" denotes the LLaMA 3.1 8B Instruct model. We make the number bold if it is the best among the four, we underline the number if it is at the second place. $^{†}$ indicates the performance reported by Gou et al. (2023); Srivastava et al. (2024); Zhang et al. (2024a).

| Models | Dev | | | | Test | | | | | | | |
|---|---|---|---|---|---|---|---|---|---|---|---|---|
| | FeT | HiT | TaF | FEV | W-T | W-S | Hyb | TAT | AIT | TAB | Inf | KVR | ToT |
| GPT-3.5 | 26.49$^{†}$ | 43.62$^{†}$ | 67.41$^{†}$ | 60.79$^{†}$ | 53.13$^{†}$ | 41.91$^{†}$ | 40.22$^{†}$ | 31.38$^{†}$ | 84.13 | 46.30$^{†}$ | 56.00 | 54.56$^{†}$ | 16.81$^{†}$ |
| GPT-4 | 21.70$^{†}$ | 48.40$^{†}$ | 74.40$^{†}$ | 71.60$^{†}$ | 68.40$^{†}$ | 47.60$^{†}$ | 58.60$^{†}$ | 55.81$^{†}$ | 88.57 | 67.10$^{†}$ | 58.60 | 56.46$^{†}$ | 12.21$^{†}$ |
| base | 15.33 | 32.83 | 58.44 | 66.37 | 43.46 | 20.43 | 32.83 | 26.70 | 82.54 | 39.97 | 48.39 | 50.80 | 13.24 |
| *TAMA* | 35.37 | 63.51 | 73.82 | 77.39 | 52.88 | 68.31 | 60.86 | 48.47 | 89.21 | 65.09 | 64.54 | 43.94 | 37.94 |

we test our model on the two synthesized table understanding datasets from Table-Syn (Li et al., 2023) and from Wu et al. (2024) (denoted as S1 and S2 in Table 7, respectively) to assess its out-of-domain table understanding ability. To assess the model's general ability, apart from reporting the model' scores on the MMLU and IFEval, we test our model on MMLU$_{Pro}$, AI2ARC, and GPQA introduced in Table 2.

Appendix A provides more details of our experimental setup including the information of GPU server, generation hyperparameters, data processing, and our evaluation setup. Appendix F provides examples from datasets that we evaluate upon.

## 4.2 RESULTS AND ANALYSIS

Table 5 shows *TAMA*'s performance on datasets listed in Table 4. Table 7 shows *TAMA*'s performance on the two out-of-domain table benchmarks and *TAMA*'s performance on the general benchmarks.

**_TAMA_ demonstrates strong table understanding ability.** We notice that there is a significant performance boost for *TAMA* compared to its base model, LLaMA 3.1 8B Instruct, on almost every dataset. For instance, on Table QA tasks such as HybridQA, *TAMA* achieves an accuracy of 60.86 compared to LLaMA 3.1 8B Instruct's 32.83. When compared to the commercial closed-source LLMs such as GPT-3.5 and GPT-4, *TAMA* surpasses the performance of GPT-3.5 model on almost

---

[1] https://machinelearning.apple.com/research/introducing-apple-foundation-models

[2] Due to budget limit for prompting GPT models, we uniformly sample 500 data points from the original test set as our test set.

Table 7: Evaluation results on the out-of-domain table understanding benchmarks and general benchmarks. For the two out-of-domain table understanding datasets, we make the number bold if it is the best among the four, we underline the number if it is at the second place. $^\dagger$ indicates results reported by Achiam et al. (2023); Zhou et al. (2023); Rein et al. (2023); Wang et al. (2024); Wu et al. (2024), and the report from Apple[1].

| Models | Out-of-Domain Table | | General | | | | |
| | Test | | Dev | | Test | | |
| | S1$^2$ (2023) | S2 (2024) | MMLU | IFEval | MMLU$_{Pro}$ | GPQA | AI2ARC |
| | Acc | ROUGE-L | Acc | Acc | Acc | Acc | Acc |
| GPT-3.5 | 54.80 | 27.75$^\dagger$ | 70.00$^\dagger$ | 74.80$^\dagger$ | - | 29.80$^\dagger$ | - |
| GPT-4 | **80.20** | **40.38**$^\dagger$ | 86.40$^\dagger$ | 92.00$^\dagger$ | 63.71$^\dagger$ | 32.10$^\dagger$ | - |
| base | 53.60 | 23.47$^\dagger$ | 66.04 | 79.62 | 22.10 | 32.14 | 80.89 |
| *TAMA* | 64.93 | 28.60 | 66.99 | 74.70 | 31.84 | 31.92 | 81.23 |

every table task in Table 5 except for KVRET and WikiTQ. And on WikiTQ, the two yields a similar performance (*TAMA* achieves 52.81 and GPT-3.5 achieves 53.13).

On tasks such as WikiSQL, HybridQA, InfoTabs, FEVEROUS, *TAMA* yields a superior performance than GPT-4. Notably, on two out-of-domain synthesized table understanding datasets in Table 7, *TAMA* surpasses the performance of GPT-3.5 (on S1, *TAMA* yields 64.93 while GPT-3.5 yields 54.80, on S2, *TAMA* yields 28.60 while GPT-3.5 yields 27.75). These two datasets are comprised of diverse table understanding tasks, and the domain distribution is significantly different from all the in-domain training data we use. The competitive performance *TAMA* demonstrates on these two datasets indicates its strong general table understanding ability. This suggests that while pre-training imparts a foundational understanding of table-related knowledge, table-specific fine-tuning plays a crucial role in further enhancing the model's capability in handling table data.

*TAMA* **preserves the general capabilities.**
In Table 7, we note that *TAMA* preserves the original LLaMA 3.1 8B Instruct's performance on almost every general benchmark. For instance, on MMLU, *TAMA* yields an accuracy of 66.99 compared to the base model's 66.04; on AI2ARC, *TAMA* yields an accuracy of 81.23 compared to the base model's 80.89. We leave the discussion of the slight performance improvements on these general benchmarks to Section 4.3. On IFEval, *TAMA* preserves most of its instruction following ability compared to the base model (74.70 compared to the base model's 79.62). Thanks to the strong instruction following ability of the original LLaMA 3.1 8B Instruct model, *TAMA* even yields a similar instruction following score on IFEval to GPT-3.5 (74.70 for *TAMA* compared to 74.80 for GPT-3.5). Table 6 provides two examples from *TAMA*'s predictions versus existing table LLMs' on IFEval and Table-Syn (S1 in Table 5). Existing table LLMs fail to return their

Table 6: Table LLMs' predictions on the prompts from IFEval and Table-Syn (S1 in Table 5). We omit parts of the examples for readability. Appendix E provides the complete examples.

| | PROMPT: | Please provide ... in **JSON format**. | Correct? |
| --- | --- | --- | --- |
| TableLLaMA | | <Mommy>, <Dad> ...  | ✗ |
| TableLLM | | ...df = pd.read_csv('data.csv')... | ✗ |
| TableBenchLLM | | ...1. Sarah Palin... | ✗ |
| *TAMA* (ours) | | {"famous_moms": [{"name": ... } | ✓ |
| | PROMPT: | # Task Description: determine the semantic type ... Return in JSON format... [Table] [Candidates]... | Correct? |
| TableLLaMA | | <Blue Blazer (mask)>,... | ✗ |
| TableLLM | | {"chosen_semantic_type": "Film"} | ✗ |
| TableBenchLLM | | ...Loser (wager)*Let's consider... | ✗ |
| *TAMA* (ours) | | {"chosen_semantic_type": "Wrestler"} | ✓ |

answers in JSON formats in most cases, while *TAMA* successfully returns the correct format.

*TAMA* **is data efficient.** We highlight that for each dataset, we use 200 training instances, which is less than 5% of the size of the original training dataset. For instance, on HiTab, we use 2.67% of the original 7,417 training instances, and on TabFact, we use 0.21% of the original 92,283 training instances. In total, we use 2,600 table instruction-answer pairs. When tuned on such a limited

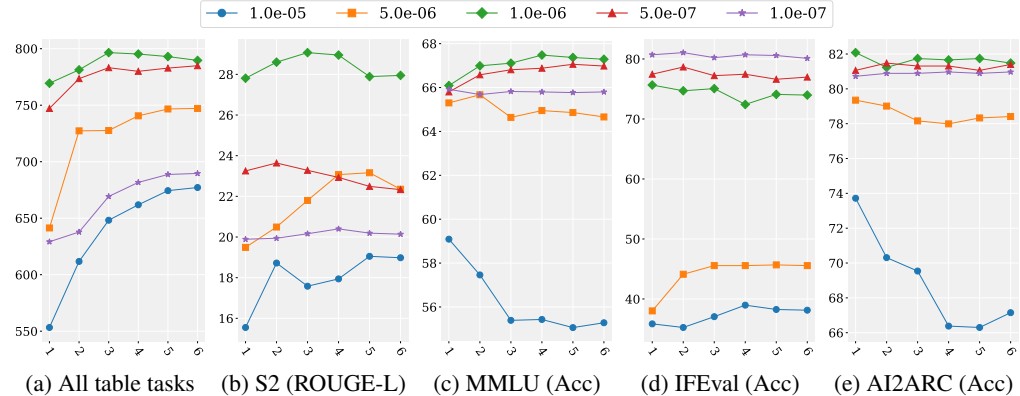

(a) All table tasks    (b) S2 (ROUGE-L)    (c) MMLU (Acc)    (d) IFEval (Acc)    (e) AI2ARC (Acc)

Figure 5: Performance scores (y-axis) with respect to the number of epochs (x-axis) and learning rates. In Figure 5a, we aggregate the performance scores for all the datasets listed in Table 4.

number of training instances, with carefully selected hyperparameters, the model can still advance its table understanding ability while maintaining its general capabilities.

### 4.3 HINDSIGHT ANALYSIS

In hindsight, we want to validate that our selected hyperparameters indeed work the best. Therefore, we run the experiments on the same training set with the learning rate ranging from 1.0e-7 to 1.0e-5, and the number of epochs from one to six. Figure 5 reports part of the results, and Figure 9 in Appendix D reports the complete results and provide further discussion.

As shown in Figure 5a, on the table understanding tasks, the learning rate of 1.0e-6 and 5.0e-7 yield the best overall performance, which coincides with our findings in Section 3. In addition, the model achieves its best aggregated performance around two to three epochs for both learning rate.

Table 8: Performance breakdown in terms of the four categories in the MMLU benchmark. The performance corresponds to the learning rate of 1.0e-6 and two training epochs.

|  | STEM | Social Science | Human- ities | Others | Overall |
|---|---|---|---|---|---|
| base | 56.03 | 76.15 | 61.57 | 72.27 | 66.04 |
| *TAMA* | 58.25 | 76.37 | 62.42 | 72.86 | 66.99 |

On S2, one of the out-of-domain table understanding datasets, the learning rate of 1.0e-6 maintains an overall best ROUGE-L score (around 28 to 29), and the learning rate of 5.0e-7 underperforms 1.0e-6, with the best ROUGE-L score of 23.64 achieved at the second epoch.

For MMLU, both 1.0e-6 and 5.0e-7 maintains their performance, sometimes even slightly better than the original LLaMA 3.1 8B Instruct model. As revealed in Table 8, the performance boost is most pronounced on STEM category. We hypothesize that this is because table-related tasks typically involves data analysis that requires math reasoning, which belongs to the STEM category. Therefore, training on table-related tasks would lead to better STEM performance. This also explains the performance boost for MMLU$_{\text{Pro}}$ in Table 7.

For IFEval, AI2ARC, the smaller the learning rate is, the less it affects the model's general capabilities. For instance, on IFEval, at the smallest learning rate of 1.0e-7, the model maintains the base model's performance, while 5.0e-7 and 1.0e-6 maintain most of the base model's performance.

Generally, the trends we observe here follow the trends we have observed in Section 3. A learning rate that is too large or too small would lead to suboptimal performance on table understanding tasks, and fine-tuning the model with one or two epochs would result in a competitive model without the risk of sacrificing its general capabilities. Moreover, we demonstrate here that with preliminary experiments, we can find a set of good or even the best hyperparameters to train the final model. Therefore, we highly recommend researchers to be mindful about the hyperparameter selection and conduct preliminary experiments when they start building their own models.

## 5  RELATED WORKS

**Table-Related Tasks.**    Tasks involving structured data, especially in the form of tables, have attracted interests from diverse communities (Deng et al., 2022a; Chen et al., 2022; Deng et al., 2022b). These tasks address diverse applications with different input-output formats. For instance, table question answering (Table QA) answers the question given the table, either in the form of natural sentences (Nan et al., 2022) or concise responses such as entities mentioned in the table, or numbers (Pasupat & Liang, 2015; Zhong et al., 2017). Table fact verification verifies a claim given the table (Chen et al., 2019; Gupta et al., 2020). Dialogue generation generates the response to the end user given the table, and the dialogue history information (Eric & Manning, 2017) Table-to-text generates a description based on the table content (Parikh et al., 2020).

**Table Understanding Methods.**    The past decade has witnessed a paradigm shift in approaches to table understanding. Before the advent of LLMs, researchers typically adapt model structures to better interpret table data (Lebret et al., 2016; Liu et al., 2018; Yang et al., 2022). As language models demonstrate promising performance on various tasks (Devlin et al., 2019), researchers gradually shift their attention towards data-driven methods for table understanding. For instance, Yin et al. (2020); Herzig et al. (2020) pre-train BERT (Devlin et al., 2019) or BERT-derived model on large-volume of table data from sources such as Wikipedia to acquire better table representations. Xie et al. (2022) reveal the synergy effects of various structured tasks, including many table tasks, laying foundations to build a generalist model for structured data. In the era of LLMs, as LLMs possess innate table-understanding abilities, researchers also explore prompt engineering techniques to optimize LLMs for table tasks (Chang & Fosler-Lussier, 2023; Deng et al., 2024).

**Table Instruction Tuning.**    Building on the advances in data-driven methods, researchers have increasingly focused on instruction tuning to enhance LLMs' table understanding ability. As demonstrated by Touvron et al. (2023); Dubey et al. (2024); Chung et al. (2024), instruction-tuning can improve model performance and generalization to unseen tasks. Meanwhile, models from the open-source LLaMA family (Touvron et al., 2023) demonstrate strong capabilities, leading researchers to instruction-tune these models for better table understanding. For instance, TableLLaMA (Zhang et al., 2024a) is instruction-tuned from a variant of LLaMA 2 model (Touvron et al., 2023), TableLLM (Zhang et al., 2024b) is instruction-tuned from CodeLLaMA, Wu et al. (2024) instruction-tune various foundational models such as LLaMA 3.1 (Dubey et al., 2024), resulting in their TableBenchLLM model. Moreover, Zheng et al. (2024) treat tables as images and instruction-tune Vicuna (Chiang et al., 2023), a vision model that is originally fine-tuned from the LLaMA model, for table understanding. However, as revealed by Zheng et al. (2024); Deng et al. (2024), treating tables as texts rather than images yields better performance. In this paper, we focus on table instruction tuning with tables fed as texts.

## 6  CONCLUSION

In this paper, we reveal the limited out-of-domain table understanding ability and general capabilities of the existing table LLMs. From our analysis, we find that the commonly-adopted hyperparameters in existing table LLMs are suboptimal, and hyperparameter choices in table instruction tuning are crucial in shaping the model's capabilities. We select hyperparameters from our analysis, and fine-tune our own model, *TAMA*. Notably, as an 8B model, *TAMA* demonstrates strong table understanding ability, outperforming GPT-3.5 on most of the table understanding benchmarks, even achieving performance on par or better than GPT-4. Moreover, *TAMA* preserves strong general capabilities. We hope our findings as well as our model *TAMA* can facilitate future research on structured data.

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

Table 9: Temperature and top_p value for table LLMs.

| Model | Temperature | Top_p |
|---|---|---|
| TableLLaMA | 0.6 | 0.90 |
| TableLLM | 0.8 | 0.95 |
| TableBenchLLM | 0.0 | 0.95 |
| *TAMA* (ours) | 0.01 | 0.95 |

## A  EXPERIMENT DETAILS

### A.1  GPU DETAILS

We run our experiments on 1 server node with 4 A40, each with 48 GB GPU memory, and 1 server node with 8 A100, each with 48 GB GPU memory.

### A.2  GENERATION DETAILS.

Table 9 shows the generation hyperparameters for table LLMs.

### A.3  DETAILS OF PROMPTING GPT MODELS

We prompt the GPT-3.5-turbo and GPT-4-turbo model and set the temperature to 0.

### A.4  DETAILS OF DATA PROCESSING

We follow the format of the dataset if the dataset is used by Zhang et al. (2024a). We add instructions for the datasets used by Xie et al. (2022). For datasets not used by Zhang et al. (2024a); Xie et al. (2022), we process them from their original source, and add an instruction per dataset.

### A.5  DETAILS OF EVALUATION

For datasets such as WikiTQ, TATQA, we follow their original evaluation scripts. For datasets such as WikiSQL, we follow Xie et al. (2022); Zhang et al. (2024a) to evaluate the exact match accuracy. For datasets such as ToTTo and FeTaQA, we follow Xie et al. (2022) and use the SacreBLEU loaded from the Hugging Face library to calculate the BLEU-4 score. For ToTTo, following Xie et al. (2022), we calculate the BLEU-4 score given all the references in the test set. For S2, we report the ROUGE-L following Wu et al. (2024) loaded from the Hugging Face library.

For MMLU, MMLU$_{\text{Pro}}$, AI2ARC and GPQA, our objective is to select the most appropriate completion among a set of given options based on the provided context. Following Touvron et al. (2023), we select the completion with the highest likelihood given the provided context. As we evaluate the model based on their selection of choice "A", "B", etc. We do not normalize the likelihood by the number of characters in the completion. We note that our setup for MMLU$_{\text{Pro}}$ is different from the chain-of-thought (CoT) (Wei et al., 2022) setup in the original LLaMA 3.1 report, as many of the existing table LLMs exhibit poor instruction-following ability, making it challenging to evaluate their performance through generation-based tasks. For IFEval, we report the instance-level strict accuracy defined by Zhou et al. (2023), which reports the percentage of verifiable instructions that are followed.

## B  EVALUATION OF THE EXISTING TABLE LLMS.

**MMLU Performance Breakdown in Terms of Categories.**  We provide the performance breakdown in terms of the category for MMLU in Table 10.

On STEM subjects, TableLLaMA experiences a decline of 7.05, while TableLLM and TableBench-LLM drop by 5.40 and 7.36, respectively. STEM subjects, including abstract algebra and mathematics at various levels (elementary, high school, and college), typically require strong logical

Table 10: Performance (accuracy scores) comparison between existing table LLMs (second row) and their base models (first row) with respect to the four categories in MMLU (e.g. "STEM" column) and their overall MMLU performance ("Overall" column). †: A variant of LLaMA 2 7B model.

| | STEM | Social Science | Humanities | Others | Overall |
|---|---|---|---|---|---|
| LongLoRA 7B[†] | 35.65 | 50.70 | 40.66 | 51.20 | 44.22 |
| TableLLaMA | 28.60 | 31.49 | 29.59 | 31.65 | 30.27 |
| Δ | ↓ 7.05 | ↓ 19.21 | ↓ 11.07 | ↓ 19.55 | ↓ 13.95 |
| CodeLLaMA 13B Instruct | 37.57 | 50.24 | 42.64 | 49.05 | 44.69 |
| TableLLM | 32.17 | 39.52 | 34.77 | 37.57 | 35.90 |
| Δ | ↓ 5.40 | ↓ 10.72 | ↓ 7.87 | ↓ 11.48 | ↓ 8.79 |
| LLaMA 3.1-8B | 52.85 | 73.94 | 55.43 | 69.06 | 62.08 |
| TableBenchLLM | 45.49 | 62.56 | 46.18 | 59.38 | 52.67 |
| Δ | ↓ 7.36 | ↓ 11.38 | ↓ 9.25 | ↓ 9.68 | ↓ 9.41 |

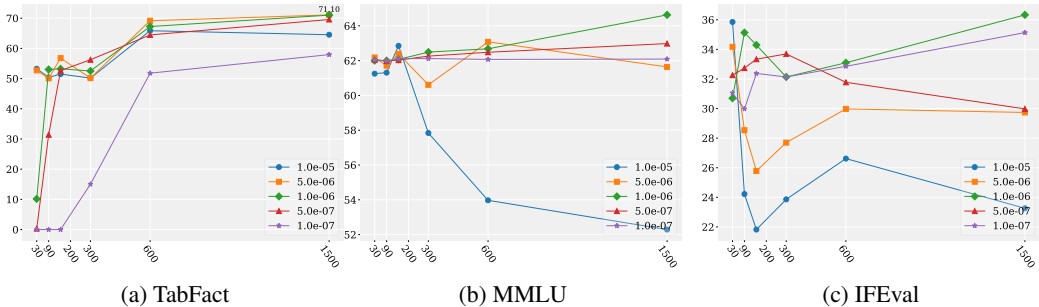

(a) TabFact            (b) MMLU            (c) IFEval

Figure 6: LLaMA 3.1 8B's accuracy scores (y-axis) on TabFact, MMLU, and IFEval with respect to the number of training instances (x-axis). We fine-tune the model for three epochs.

reasoning and analytical capabilities, which are highly relevant to data analysis in table tasks. The drop in performance across these models indicates that current table instruction tuning compromises such reasoning abilities of their base models, limiting their application in table analytical scenarios.

There is even more pronounced performance degradation in other categories. Though these categories may not directly align with table understanding, they assess model capabilities that are still critical for end-user applications. For instance, the "Others" category includes subjects like global facts, which are essential for users seeking reliable information during queries. The decline in performance across these broader categories suggests that the current table instruction tuning methods may compromise the model's ability to handle general knowledge tasks effectively, which limits its practical usefulness for diverse real-world applications.

## C  MODEL AND HYPERPARAMETER EXPLORATION

### C.1  MODEL SELECTION

**Reasons to Select LLaMA 3.1.**  LLaMA 3.1 (Dubey et al., 2024) provides a set of foundational models for language. Compared to the prior LLaMA models, LLaMA 3.1 claims to improve both the quantity and the quality of the data used for pre-training and post-training (15T multilingual pre-training tokens for LLaMA 3.1 compared to 1.8T tokens for LLaMA 2). Such an enormous amount of training makes LLaMA 3.1 one of the most advanced open-source LLMs.

**Reasons to Select the Instruct Version Rather than the Base Version.**  Currently, there are two kinds of model selections for table instruction tuning, instruction-tuning the base version of the model, as seen in works like TableLLaMA(Zhang et al., 2024a) and TableBenchLLM(Wu

et al., 2024), or continuing instruction-tuning an already instruction-tuned version, as done with TableLLM(Zhang et al., 2024b) as listed in Table 1.

As the end user may come up with their own set of instructions, we expect table instruction-tuned models to possess a strong general instruction-following ability. Imparting general instruction-following ability through table instruction-tuning to the base model is challenging, as there is a lack of diversity in the table instruction-tuning data. For instance, TableLLaMA employs six specific instruction templates across two million data points, which pales in comparison to the diverse instruction datasets in broader instruction tuning efforts such as those by Chung et al. (2024), which include 1,836 tasks, each with a set of instruction templates. As shown in Figure 6c, when tuning the base version of the LLaMA 3.1 8B model on instruction pairs on FeTaQA, HiTab, and TabFact, the instruction following ability of the model does not improve significantly. Moreover, with a large learning rate such as 1.0e-5, the model's instruction following ability drops significantly when there is more training data coming in.

We argue that the instruction-tuned version possesses strong general instruction-following capabilities, eliminating the need to repeat the general instruction-tuning stage. Therefore, *a more effective strategy is to table instruction-tune an already instruction-tuned model, focusing on enhancing its table understanding ability while preserving its general instruction-following capabilities*. As shown in Figure 2f, with proper hyperparameter selection, we can maintain the inherent strong instruction following ability of the LLaMA 3.1 8B Instruct model.

In terms of specific table understanding tasks, tuning LLaMA 3.1 8B Instruct model yields better performance than its base version on TabFact (73.10 in Figure 2a v.s. 71.10 in Figure 6a) under the same experimental setup. Therefore, we select the LLaMA 3.1 8B Instruct model as our starting model.

## C.2 Hyperparameter Exploration Across Models

We conduct experiments to validate our findings across different models in the full-parameter setup, including Llama 2 7B Instruct (Touvron et al., 2023), QWen 2.5 7B Instruct (Bai et al., 2023), Mistral v0.3 7B Instruct (Jiang et al., 2023), and Phi 3 small 8K Instruct (7B) (Abdin et al., 2024).

**Learning Rate.** We train each model on 500 examples from HiTab, FeTaQA, and TabFact (1,500 examples total) to explore the effects of the learning rate. Table 11 presents our results.

Table 13: Recommended learning rate across different LLMs on table-specific tasks.

| Model | Learning Rate |
|---|---|
| Llama 2 7B Instruct | 1.0e-6 / 5.0e-7 |
| Llama 3.1 8B Instruct | 1.0e-6 / 5.0e-7 |
| QWen 2.5 7B Instruct | 1.0e-6 / 5.0e-7 |
| Mistral v0.3 7B Instruct | 5.0e-7 / 1.0e-7 |
| Phi 3 small 8K Instruct (7B) | 5.0e-6 / 1.0e-6 |

We observe a significant performance drop happens for every model on the two general benchmarks. Interestingly, for models such as QWen 2.5, when we increase the learning rate from 1.0e-6 to 5.0e-6, it would primarily affect the IFEval dataset rather than MMLU, suggesting that the compromises may happen at different speeds with respect to different aspects of the model's general capability.

The Phi model shows a pronounced performance drop from 1.0e-5 to 5.0e-5, in contrast to Llama, Mistral and QWen models, where the "breakdown point" on the learning rate is slightly smaller, especially for Mistral model, where we see 5 points lose on IFEval from 5.0e-7 to 1.0e-6.

Table 13 lists the learning rate we would suggest for practitioners to use if they would fine-tune the LLMs on table-specific tasks.

**Number of Examples.** We further experiment with various training sizes for each model to observe its impact on performance. Table 12 reports the results for Llama 2 7B, QWen 2.5, Mistral v0.3, and Phi 3 8K models at one of the learning rates we select based on our results in Table 11.

Across all models, performance improvement becomes marginal from 600 to 1500 examples, suggesting diminishing returns with larger datasets.

Table 11: LLMs' performance scores corresponding to different learning rate. In this experiment, we train each model on 500 examples from HiTab, FeTaQA, and TabFact (1,500 examples total) for three epochs.

| Learning Rate | FeTaQA | TabFact | MMLU | IFEval |
|---|---|---|---|---|
| *Llama 2 7B Instruct* | | | | |
| 5.0e-7 | 26.54 | 52.63 | 47.12 | 47.84 |
| 1.0e-6 | 29.03 | 53.80 | 47.07 | **47.84** |
| 5.0e-6 | 33.86 | 51.05 | 46.58 | **35.25** |
| 1.0e-5 | 34.77 | 53.79 | 45.99 | 39.93 |
| *QWen 2.5 7B Instruct* | | | | |
| 5.0e-7 | 33.14 | 71.09 | 73.66 | 76.02 |
| 1.0e-6 | 34.50 | 72.66 | 73.52 | **75.78** |
| 5.0e-6 | 34.04 | 72.81 | 73.81 | **49.28** |
| 1.0e-5 | 33.84 | 71.51 | 73.49 | 41.61 |
| *Mistral v0.3 7B Instruct* | | | | |
| 1.0e-7 | 31.91 | 64.32 | 61.32 | 62.83 |
| 5.0e-7 | 36.44 | 70.35 | 60.76 | 57.79 |
| 1.0e-6 | 36.99 | 71.88 | 60.45 | **52.28** |
| 5.0e-6 | 35.71 | 53.64 | 34.96 | **33.09** |
| 1.0e-5 | 32.14 | 50.87 | 24.93 | 27.70 |
| *Phi 3 8K Instruct (7B)* | | | | |
| 1.0e-6 | 33.10 | 72.04 | 70.48 | 71.22 |
| 5.0e-6 | 37.26 | 73.82 | 74.89 | 68.71 |
| 1.0e-5 | 38.13 | 73.92 | 73.30 | **62.95** |
| 5.0e-5 | 34.46 | 50.90 | 49.08 | **28.78** |
| 1.0e-4 | 30.66 | 50.33 | 49.17 | 23.02 |

Table 12: LLMs' performance scores corresponding to different sizes of the training data. We specify the learning rate we use for each model in the bracket next to the model names. Here we train each model for three epochs.

| # Size | FeTaQA | TabFact | MMLU | IFEval |
|---|---|---|---|---|
| *Llama 2 7B Instruct (1.0e-6)* | | | | |
| 30 | 13.32 | 31.68 | 47.07 | 45.08 |
| 90 | 13.86 | 49.51 | 46.96 | 46.16 |
| 150 | 14.79 | 46.24 | 47.09 | 47.48 |
| 300 | 14.47 | 50.27 | 47.09 | 45.56 |
| 600 | 24.12 | 50.74 | 47.11 | 45.56 |
| 1500 | 29.03 | 53.80 | 47.07 | 47.84 |
| *QWen 2.5 7B Instruct (1.0e-6)* | | | | |
| 30 | 14.2 | 8.42 | 73.91 | 70.43 |
| 90 | 16.45 | 8.47 | 73.76 | 70.43 |
| 150 | 21.14 | 69.66 | 73.83 | 69.5 |
| 300 | 22.1 | 69.65 | 73.72 | 68.95 |
| 600 | 32.12 | 70.86 | 73.71 | 68.21 |
| 1500 | 34.5 | 72.66 | 73.52 | 66.73 |
| *Mistral v0.3 7B Instruct (5.0e-7)* | | | | |
| 30 | 23.84 | 0.28 | 61.39 | 49.72 |
| 90 | 10.67 | 60.29 | 61.34 | 51.76 |
| 150 | 19.79 | 49.82 | 61.34 | 52.87 |
| 300 | 33.93 | 61.91 | 61.13 | 51.02 |
| 600 | 34.28 | 66.34 | 61.12 | 52.31 |
| 1500 | 36.44 | 70.35 | 60.76 | 47.69 |
| *Phi 3 8K Instruct (7B) (5.0e-6)* | | | | |
| 30 | 17.19 | 9.62 | 75.43 | 52.31 |
| 90 | 24.01 | 67.32 | 75.43 | 63.96 |
| 150 | 24.67 | 68.00 | 75.43 | 62.11 |
| 300 | 34.81 | 71.30 | 75.61 | 62.85 |
| 600 | 37.74 | 72.91 | 75.50 | 61.18 |
| 1500 | 37.26 | 73.82 | 75.26 | 59.70 |

In addition, we find that given the same number of training instances, Llama 3.1 8B Instruct achieves better performance than Llama 2 7B Instruct. For instance, when trained with the same 1,500 examples at the learning rate of 1.0e-6, Llama 3.1 8B Instruct yields 73.10 on TabFact (Section 3) while Llama 2 7B Instruct only yields 53.80 (Table 12). Therefore, models with stronger general capabilities require less tuning data in our fine-tuning process.

## C.3 Hyperparameter Exploration for LoRA and QLoRA

We conduct experiments using LoRA (Hu et al., 2021) and QLoRA (Dettmers et al., 2024) based on Llama 3.1-8B-Instruct. Specifically, we use hugging-quants/Meta-Llama-3.1-8B-Instruct-AWQ-INT4 [1] as the base model for our QLoRA experiments.

We replicate the experiments we conduct in Appendix C.2, and here we present our results in two aspects, the learning rate and the number of examples.

---

[1] `https://huggingface.co/hugging-quants/Meta-Llama-3.1-8B-Instruct-AWQ-INT4`

Table 14: Performance scores corresponding to using LoRA and QLoRA. In this experiment, we train each model on 500 examples from HiTab, FeTaQA, and TabFact (1,500 examples total) for three epochs.

| Learning Rate | FeTaQA | TabFact | MMLU | IFEval |
|---|---|---|---|---|
| *LoRA* | | | | |
| 1.0e-6 | 16.63 | 63.21 | 66.06 | 80.22 |
| 5.0e-6 | 23.69 | 66.80 | 65.97 | 80.94 |
| 1.0e-5 | 29.66 | 68.58 | 66.03 | **80.58** |
| 5.0e-5 | 35.33 | 73.80 | 67.04 | **76.98** |
| 1.0e-4 | 35.81 | 75.63 | 67.42 | **71.22** |
| 5.0e-4 | 36.04 | 73.88 | 66.36 | **60.67** |
| 1.0e-3 | 35.54 | 73.64 | 59.02 | 38.73 |
| *QLoRA* | | | | |
| 1.0e-7 | 20.36 | 63.06 | 64.56 | 80.22 |
| 5.0e-7 | 19.07 | 66.42 | 64.68 | 80.46 |
| 1.0e-6 | 27.44 | 67.18 | 64.68 | 79.98 |
| 5.0e-6 | 34.64 | 70.98 | 64.76 | 78.66 |
| 1.0e-5 | 36.86 | 73.20 | 65.22 | 77.58 |
| 5.0e-5 | 36.52 | 74.11 | 65.82 | 76.02 |
| 1.0e-4 | 35.94 | 74.91 | 65.76 | **74.22** |
| 5.0e-4 | 33.72 | 50.50 | 42.76 | **32.85** |
| 1.0e-3 | 0.01 | 50.16 | 22.95 | 23.86 |

Table 15: Performance scores corresponding to different sizes of the training data for LoRA and QLoRA. We specify the learning rate we use for LoRA and QLoRA in the bracket next to the method names.

| # Size | FeTaQA | TabFact | MMLU | IFEval |
|---|---|---|---|---|
| *LoRA (5.0e-5)* | | | | |
| 30 | 17.36 | 63.89 | 66.14 | 71.90 |
| 90 | 19.83 | 66.50 | 66.03 | 70.98 |
| 150 | 14.69 | 68.62 | 66.10 | 73.01 |
| 300 | 26.01 | 67.96 | 66.20 | 72.09 |
| 600 | 34.08 | 72.13 | 66.65 | 70.61 |
| 1500 | 35.33 | 73.80 | 67.04 | 68.39 |
| *QLoRA (5.0e-5)* | | | | |
| 30 | 18.02 | 66.55 | 64.78 | 72.46 |
| 90 | 35.33 | 68.44 | 65.08 | 69.32 |
| 150 | 33.50 | 69.78 | 65.36 | 74.31 |
| 300 | 35.95 | 69.46 | 65.63 | 71.72 |
| 600 | 36.25 | 73.68 | 65.80 | 69.13 |
| 1500 | 36.52 | 74.11 | 65.82 | 65.62 |

**Learning Rate.** Table 14 presents the results. We find that there is still a "breakdown point" where further increasing the learning rate causes a sharp decline in overall performance for both LoRA and QLoRA. However, such "breakdown point" for LoRA and QLoRA (around 5.0e-5) is larger than the full parameter tuning (usually around 1.0e-6). When the learning rate does not surpass such a "breakdown point", both methods demonstrate competitive in-domain performance on table tasks.

**Number of Examples.** Table 15 presents the results. Similar to what we have found for full parameter fine-tuning, both LoRA and QLoRA show diminishing returns as the number of training examples increases. While performance improves with more examples, the rate of improvement slows beyond 600 examples for LoRA. For QLoRA, the rate of improvement slows beyond 90 examples. We find that with 1,500 examples, QLoRA and LoRA perform similarly on the in-domain table tasks, and on FeTaQA, QLoRA even outperforms LoRA by 1 point. This suggests that practitioners may leverage such parameter-efficient fine-tuning methods like QLoRA in practice, especially when they have limited table data.

## C.4 INDIVIDUAL TASK'S INFLUENCE ON MODEL PERFORMANCE

Figures 7 and 8 present heatmaps across varying learning rates (from 1.0e-7 to 1.0e-5) and epochs (from one to six). We can see that the patterns coincide with what we have discussed in Section 3, that a learning rate that is too large such as 1.0e-5 or too small such as 1.0e-7 leads to suboptimal table understanding ability, and the large learning rate also compromises the model's general capabilities. Moreover, we do not observe significant performance gain when we fine-tune the model for more epochs. Across these hyperparameters, we can observe the inter-connections between tasks such as HiTab and TabFact, as training solely on one often leads to good performance on the other. But this is not universally true, as tasks such as FeTaQA and FEVEROUS seem to not have strong inter-connections.

In addition, we observe that *the learning rate works the best for an individual task does not necessarily work the best for other tasks*. For instance, in Figures 7 and 8, the learning rate of 5.0e-6 yields the best performance for FeTaQA, but is suboptimal for HiTab and TabFact. This highlights

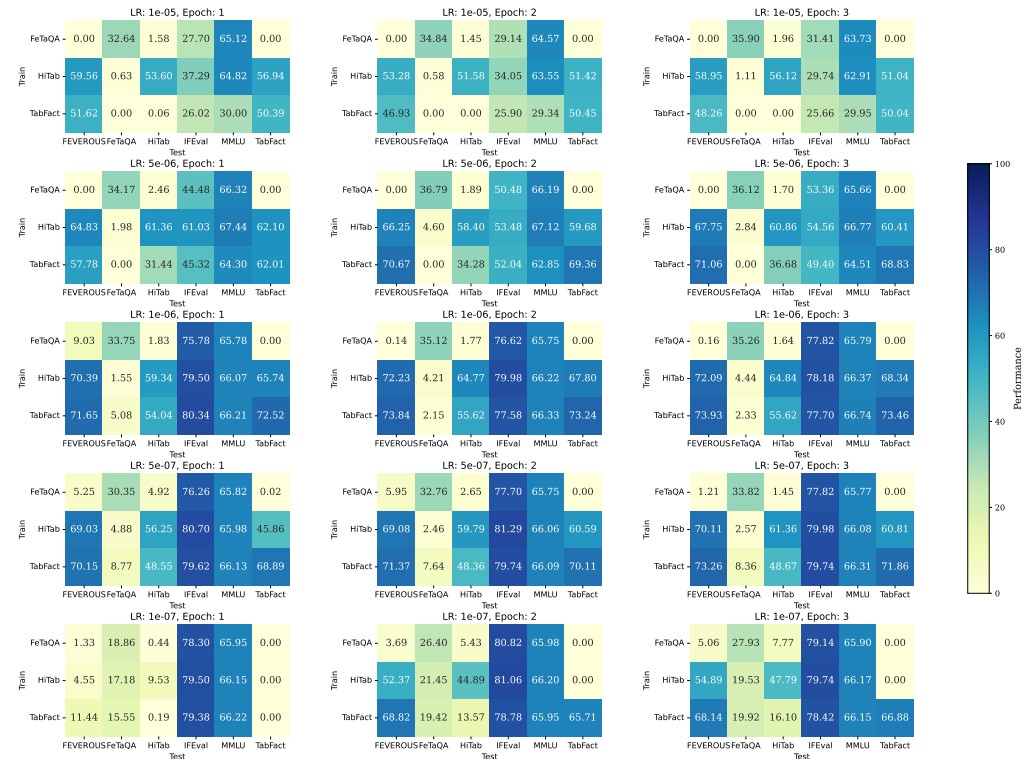

Figure 7: Heatmap when we fine-tune LLaMA 3.1 8B Instruct model on a single dataset (y-axis) and test against the others (x-axis). We fine-tune the model for one to three epochs (horizontal directions) at a learning rate of 1.0e-5, 5.0e-6, 1.0e-6, 5.0e-7, 1.0e-7 (vertical direction) with 500 instruction pairs.

Table 16: Llama 3 8B Instruct's performance on the general benchmarks MMLU and IFEval corresponding to different learning rates (Numbers in the bracket). We train the model for three epochs using 500 examples on each dataset, respectively. "$D_{num, tab}$" represents the density of the number cells in the table. "No. Cells : Num. Cells" denotes the cells containing no number versus cells containing numbers. "TT tokens", "Tab tokens", "Q tokens" represent the total number of input tokens, table tokens, and question tokens.

| | $D_{num, tab}$ (%) | No. Cells : Num. Cells | TT tokens | Tab tokens | Q tokens | Tab tokens : Q tokens | MMLU (1e-6) | MMLU (5e-6) | MMLU (1e-5) | IFEval (1e-6) | IFEval (5e-6) | IFEval (1e-5) |
|---|---|---|---|---|---|---|---|---|---|---|---|---|
| TabFact | 73.03 | 1.34 : 1 | 292,822 | 264,520 | 19,286 | 13.72 : 1 | 66.74 | **64.51** | **29.95** | 77.70 | **49.40** | **25.66** |
| FeTaQA | 57.99 | 1.68 : 1 | 309,624 | 251,697 | 42,492 | 5.92 : 1 | 65.79 | 65.66 | 63.73 | 77.82 | 53.36 | 31.41 |
| HiTab | 80.60 | 1.19 : 1 | 452,149 | 424,941 | 11,030 | 38.53 : 1 | 66.37 | 66.77 | 62.91 | 78.18 | 49.40 | 29.74 |

that when multiple tasks are involved in the training process, researchers need to consider beyond a single task to decide their hyperparameters.

## C.5 TRADE-OFF ANALYSIS FOR DATA PROPERTIES

We expand our analysis to assess how features in the training data may influence model performance. To investigate this, we train the Llama 3.1 8B Instruct model for three epochs using 500 examples on each dataset, respectively.

Table 16 presents the results. We find that the performance degradation is most significant on TabFact. Interestingly, despite TabFact having intermediate numeric density and table-to-question token ratios, it still shows the fastest performance decline.

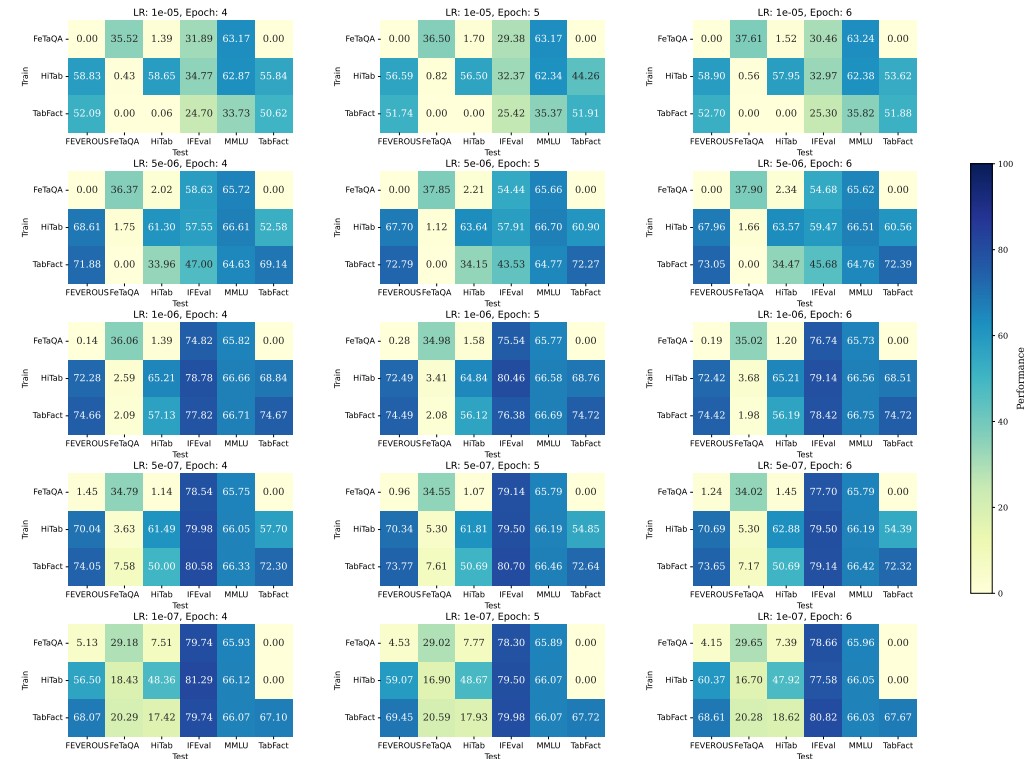

Figure 8: Heatmap when we fine-tune LLaMA 3.1 8B Instruct model on a single dataset (y-axis) and test against the others (x-axis). We fine-tune the model for four to six epochs (horizontal directions) at a learning rate of 1.0e-5, 5.0e-6, 1.0e-6, 5.0e-7, 1.0e-7 (vertical direction) with 500 instruction pairs.

We hypothesize that this is due to the nature of the task rather than the table-specific features examined. Since FeTaQA and HiTab are table QA tasks, they may possess similar QA form that the model has encountered in its general instruction tuning stage, this may ease the decay of the model's general capabilities in our fine-tuning stage. However, TabFact is about fact-checking, the input form includes both the table and the claim to be verified, which we suspect may not be as common as the QA data in its general instruction tuning stage. Therefore, the model suffers a more significant performance decay because it needs to update more of its internal knowledge to handle such a task.

## D    HINDSIGHT ANALYSIS

Figure 9 provides the complete results of the model performance versus the learning rate and the number of epochs.

Apart from what we have discussed in Section 4.3, we find that on S1, the learning rate of 5.0e-7 yields a consistent good accuracy scores (around 64 to 65) across all the epochs, while 1.0e-6 maintains a good accuracy score (around 64 to 65) for the first two epochs, but starting from the third epoch, it experiences a performance decline (from 64.93 to 52.56).

In terms of the general benchmarks, GPQA resembles similar trends as the trends for IFEval and AI2ARC that the smaller the learning rate is, the less it affects the model's general capabilities.

## E    MODEL PREDICTION EXAMPLES

Table 17 provides an example for table LLMs' generation on IFEval dataset. Tables 18 and 19 provide two examples for table LLM's generation on Table-Syn dataset. Apart from the limited out-of-domain table reasoning ability, we find that existing table LLMs also exhibit limited instruction-

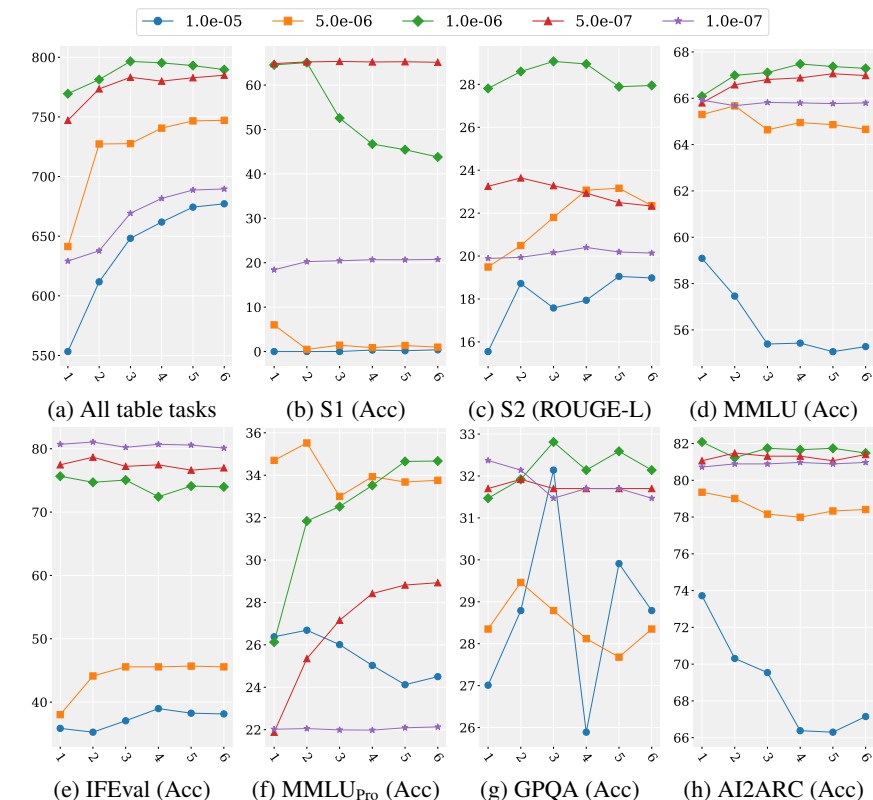

Figure 9: Performance scores (y-axis) with respect to the number of epochs (x-axis) and learning rates. In Figure 9a, we aggregate the performance scores for all the datasets listed in Table 4.

following capabilities, and often struggle with consistently returning answers in specified formats, such as JSON. Such a limitation poses challenges in the practical use cases, where the end-users may request specific output formats to extract answers from the model's predictions.

## F  DATASET EXAMPLES

### F.1  WIKITQ

**Input:**

```
[TAB] col: | description losses | 1939/40 | 1940/41 | 1941/42 | 1942/43 |
 1943/44 | 1944/45 | total | [SEP] | direct war losses | 360,000 | | | |
| 183,000 | 543,000 | [SEP] | murdered | 75,000 | 100,000 | 116,000 |
133,000 | 82,000 | | 506,000 | [SEP] | deaths in prisons & camps | 69,000
 | 210,000 | 220,000 | 266,000 | 381,000 | | 1,146,000 | [SEP] | deaths
outside of prisons & camps | | 42,000 | 71,000 | 142,000 | 218,000 | |
473,000 | [SEP] | murdered in eastern regions | | | | | | 100,000 |
100,000 | [SEP] | deaths other countries | | | | | | | 2,000 | [SEP] |
total | 504,000 | 352,000 | 407,000 | 541,000 | 681,000 | 270,000 |
2,770,000 |\n\nhow many people were murdered in 1940/41?
```

**Instruction:**

```
This is a table QA task. The goal of this task is to answer the question
given the table.
```

**Output:**

```
100,000
```

Table 17: Table LLMs' predictions on the prompt in IFEval. For TableBenchLLM, it starts to repeat itself and we omit the repeated part for better readability. †: We note that IFEval checks the instruction following ability rather than the correctness of the answer.

| | PROMPT: Please provide the names of 5 famous moms in JSON format. | Follow Instruction?[†] |
|---|---|---|
| TableLLaMA | <Mommy>, <Dad>, <Baby>, <Baby>, <Dad>, <Mommy> | ✗ |
| TableLLM | import pandas as pd
# Load the data
df = pd.read_csv('data.csv')
# Select the first 5 rows
df_first_5_rows = df.head(5)
# Convert the dataframe to JSON
json_output = df_first_5_rows.to_json()
# Print the JSON output
print(json_output) | ✗ |
| TableBenchLLM | Here are the names of 5 famous moms in JSON format: 1. Sarah Palin 2. Jill Biden 3. Michelle Obama 4. Amy Schumer 5. Madonna... | ✗ |
| *TAMA* (ours) | {"famous_moms": [{"name": "Oprah Winfrey"}, {"name": "Michelle Obama"}, {"name": "Angelina Jolie"}, {"name": "Madonna"}, {"name": "J.K. Rowling"}]]} | ✓ |

Table 18: An example of table LLMs' predictions on Table-Syn.

| | PROMPT: | Correct? |
|---|---|---|
| | Task Description: Please check the following table, there is one and exactly one cell in the table that is missing. When you find this missing cell, please point it out using the row id shown in the first column. Return the final result as JSON in the format {"row_id": "<row_id of the row with missing cell>"}.

## Input:
\|row_id\|Date\|Sales\|Cost of Sales\|Expenses\|
\|—\|—\|—\|—\|—\|
\|1\|35428\|177911\|80685\|28668\|
\|2\|\|133281\|73456\|26892\|
\|3\|36406\|164055\|88108\|28457\|
\|4\|36418\|151118\|70952\|29138\|
\|5\|37141\|168833\|77020\|23798\|

Return the final result as JSON in the format {"row_id": "<row_id of the row with missing cell>"}.

## Output: | |
| TableLLaMA | <177911.0>, <133281.0>, <164055.0>, <151118.0>, <37141.0> | ✗ |
| TableLLM | {"row_id": 2} | ✓ |
| TableBenchLLM | Final Answer: 2 // 2php2://... | ✗ |
| *TAMA* (ours) | {"row_id": 2} | ✓ |

## F.2 FETAQA

**Input:**

```
[TLE] The Wikipedia page title of this table is Gerhard Bigalk. The
Wikipedia section title of this table is Ships attacked. [TAB] | Date |
Name | Nationality | Tonnage (GRT) | Fate | [SEP] | 14 June 1941 | St.
Lindsay | United Kingdom | 5,370 | Sunk | [SEP] | 21 December 1941 | HMS
```

Table 19: An example of table LLMs' predictions on Table-Syn. We omit part of the prompt for better readability.

| | | Correct? |
|---|---|---|
| PROMPT: | # Task Description: Please look at the input column and determine the semantic type that can describe *every single* instance the input column. Please only choose one semantic type from the candidate list, and remember that the type you choose has to accurately describe every single entity in the column. If no candidate column type can suitably describe every single instance in the column, please return 'None'. Please only choose one type from the candidate list below, and *do not* create new types. Return the final result as JSON in the format {"chosen_semantic_type": "<an entry from the candidate list or None>"}.

## Input:
**Column:**
\|Loser (wager)\|
\|—\|
\|Ultratumba (mask)\|
\|Ultratumba (hair)\|
\|El Noruego (hair)\|
\|El Destroyer (mask)\|
\|Dory Dixon (hair)\|
\|Humberto Garza (hair)\|
\|CÃ©sar Valentino (hair)\|
\|La Momia (mask)\|
...
**Candidate column type:**
AcademicJournal, AdministrativeRegion, Airport...Film, Wrestler...
Return the final result as JSON in the format {"chosen_semantic_type": "<an entry from the candidate list or None>"}.
## Output: | |
| TableLLaMA | <Blue Blazer (mask)>, <Don Corleone (mask)>... | ✗ |
| TableLLM | {"chosen_semantic_type": "Film"} | ✗ |
| TableBenchLLM | ...Loser (wager)*Let's consider the new type "Loser (wager)" to be a role... | ✗ |
| *TAMA* (ours) | {"chosen_semantic_type": "Wrestler"} | ✓ |

```
Audacity | Royal Navy | 11,000 | Sunk | [SEP] | 2 February 1942 | Corilla
 | Netherlands | 8,096 | Damaged | [SEP] | 4 February 1942 | Silveray |
United Kingdom | 4,535 | Sunk | [SEP] | 7 February 1942 | Empire Sun |
United Kingdom | 6,952 | Sunk | [SEP] | 16 May 1942 | Nicarao | United
States | 1,445 | Sunk | [SEP] | 19 May 1942 | Isabela | United States |
3,110 | Sunk |\n\nThe highlighted cells of the table are: [
HIGHLIGHTED_BEGIN] [11,000], [Sunk], [8,096], [Damaged] [HIGHLIGHTED_END]
 What happened to the two heaviest ships Gerhard Bigalk attacked?
```

**Instruction:**

```
This is a free-form table question answering task. The goal for this task
 is to answer the given question based on the given table and the
highlighted cells.
```

**Output:**

```
Gerhard Bigalk damaged one ship of 8,096 GRT, and sunk one warship of
11,000 tons.
```

### F.3 TABFACT

**Input:**

[TLE] The table caption is about tony lema. [TAB] | tournament | wins | top – 5 | top – 10 | top – 25 | events | cuts made [SEP] | masters tournament | 0 | 1 | 2 | 4 | 4 | 4 | [SEP] | us open | 0 | 2 | 3 | 4 | 6 | 5 | [SEP] | the open championship | 1 | 2 | 2 | 2 | 3 | 3 | [SEP] | pga championship | 0 | 0 | 1 | 2 | 5 | 4 | [SEP] | totals | 1 | 5 | 8 | 12 | 18 | 16 |\n\nThe statement is: <tony lema be in the top 5 for the master tournament , the us open , and the open championship>. Is it entailed or refuted by the table above?

**Instruction:**

This is a table fact verification task. The goal of this task is to distinguish whether the given statement is entailed or refuted by the given table.

**Output:**

entailed

### F.4 KVRET

**Input:**

col : event | time | date | room | agenda | party\n\nThe dialogue history is: <remind me to take my pills || >. Please generate the response based on the given table and the given dialogue history.

**Instruction:**

This is a dialogue response generation task grounded on tables. The goal of this task is to generate response based on the given dialogue history and the given table. The dialogues are grounded through underlying tables and span three distinct tasks in the in-car personal assistant space: calendar scheduling, weather information retrieval, and point-of-interest navigation.

**Output:**

what time do you need to take your pills ?

### F.5 TOTTO

**Input:**

<page_title> List of Governors of South Carolina </page_title> < section_title> Governors under the Constitution of 1868 </section_title> <table> <cell> 76  #   74 </ col_header>  75  </cell> <cell> Daniel Henry Chamberlain  Governor   76 </ row_header> </cell> <cell> December 1, 1874  Took Office </ col_header>  76  </cell> </table>\n\nPlease generate one natural language description to describe the given highlighted table cells.

**Instruction:**

This is a highlighted cells description task. The goal of this task is to generate the language description given table cells.

**Output:**

Daniel Henry Chamberlain was the 76th Governor of South Carolina from 1874.

## F.6   S1

**Input:**

# Task Description: Please determine whether Citation A and Citation B refer to the same entity or not. Your final answer should be 'Yes' or 'No'. Return the final result as JSON in the format {\"answer\": \"<Yes or No>\"}. Let's think step by step and show your reasoning before showing the final result.\n\n## Input:\nCitation A is:\n|title|authors|venue|year|\n|---|---|---|---|\n|algorithms for mining distance-based outliers in large datasets|e knorr , r ng|vldb|1998|\n\nCitation B is:\n|title|authors|venue|year|\n|---|---|---|---|\n|algorithms for mining association rules in bag databases|py hsu , yl chen , cc ling|information sciences\u00e2 ?? informatics and computer science : an & hellip ;,|2004.0|\n\nLet's think step by step and show your reasoning before showing the final result. Return the final result as JSON in the format {\"answer\": \"<Yes or No>\"}.\n## Output:\n

**Instruction:**

You are a helpful assistant that specializes in tables.

**Output:**

{\"answer\": \"No\"}

## F.7   S2

**Input:**

You are a table analyst. Your task is to answer questions based on the table content.\n\n\nThe answer should follow the format below:\n[Answer Format]\nFinal Answer: AnswerName1, AnswerName2...\n\nEnsure the final answer format is the last output line and can only be in the \"Final Answer: AnswerName1, AnswerName2...\" form, no other form. Ensure the \"AnswerName\" is a number or entity name, as short as possible, without any explanation.\n\n\nGive the final answer to the question directly without any explanation.\n\nRead the table below in JSON format:\n[TABLE] \n{\"columns\": [\"season\", \"tropical lows\", \"tropical cyclones\", \"severe tropical cyclones\", \"strongest storm\"], \"data\": [[\"1990 – 91\", 10, 10, 7, \"marian\"], [\"1991 – 92\", 11, 10, 9, \"jane – irna\"], [\"1992 – 93\", 6, 3, 1, \"oliver\"], [\"1993 – 94\", 12, 11, 7, \"theodore\"], [\"1994 – 95\", 19, 9, 6, \"chloe\"], [\"1995 – 96\", 19, 14, 9, \"olivia\"], [\"1996 – 97\", 15, 14, 3, \"pancho\"], [\"1997 – 98\", 10, 9, 3, \"tiffany\"], [\"1998 – 99\", 21, 14, 9, \"gwenda\"], [\"1999 – 00\", 13, 12, 5, \"john / paul\"]]}\n\nLet's get start!\nQuestion: What is the average number of tropical cyclones per season?\n

**Instruction:**

You are a helpful assistant that specializes in tables.

**Output:**

10.6

## F.8   MMLU

**Input:**

{5-shot examples}
Find the degree for the given field extension Q(sqrt(2), sqrt(3), sqrt(18)) over Q.
\nA. 0\nB. 4\nC. 2\nD. 6\nAnswer:

**Instruction:**

```
The following are multiple choice questions (with answers) about abstract
 algebra.\n\n
```

**Output:**

```
B
```

### F.9  IFEVAL

**Input:**

```
Can you help me make an advertisement for a new product? It's a diaper
that's designed to be more comfortable for babies and I want the entire
output in JSON format.
```

**Instruction:**

```
You are a helpful assistant.
```

**Output:**

```
[JSON formatted answer]
```

