# OpenReview forum: "Rethinking Table Instruction Tuning"
_ICLR.cc/2025/Conference — Submitted to ICLR 2025_

### Official Review · Reviewer_nNeC · 2024-10-15

**Soundness:** 3
**Presentation:** 3
**Contribution:** 1
**Rating:** 5
**Confidence:** 4

**Summary:**

The paper first evaluates existing table LLMs and finds that they have poor OOD table understanding capability, poor instruction-following capability, and comprised general capabilities. The paper then studies the influence of hyperparameters on training table LLMs. Finally, the paper trains TAMA from LLaMA-3.1-8B-Instruct with great table understanding capability while preserving the general capabilities.

**Strengths:**

+ The paper illuminates the limitations/weaknesses of existing table LLMs.
+ TAMA is a good contribution to the open-source LLM community, which has a good performance. People can do follow-up work on TAMA to study LLM's specific capabilities.

**Weaknesses:**

While TAMA is a good contribution to the community, I am personally not sure about the contribution of this paper itself. The takeaway for training TAMA seems to be just a good hyperparameter choice + multi-task training. The latter is a standard way and the former does not provide too much insight - I am not saying hyperparameter tuning/search is meaningless, and my point is that the current Section 3 is more like just an experimental report, where people (at least I) might want to see some fresh insights (e.g., training dynamics, capability analysis, etc). If it is just that the previous table LLM training + careful hyperparameter selection = TAMA, I would doubt the contributions of the paper.

I am open to different opinions. If the authors can make a rebuttal that I feel is convincing, I would be happy to increase my score.

**Questions:**

N/A

---

> ### Author Response · Authors · 2024-11-19
>
> Thank you for acknowledging that our paper highlights the limitations of existing table LLMs and recognizes TAMA's value to the open-source LLM community.
>
>
> Regarding your concern
>
> > Not sure about the contribution of this paper itself.
>
> We would like to emphasize the three folds of contributions of our work.
>
> First, we systematically examine the overspecialization issues in existing table LLMs, which aligns with the observations by reviewer 8ejo04 in practice.
> Our analysis reveals the potential trade-offs between task specialization and the model’s general capabilities.
> We hope our research encourages the community to reconsider whether the performance gains claimed in state-of-the-art table LLMs justify the loss of their general capabilities.
> We are glad to see that our work has already sparked practice changes in our community – a recent paper [1] has cited our anonymized submission and discusses such trade-offs when they build their table LLM.
>
> In addition, as pointed out by reviewer 8ejo04 and 6ciM, we conduct extensive experiments to demonstrate the influence of hyperparameter settings, which we believe will help practitioners build their customized models efficiently and effectively.
> Our paper demonstrates that with significantly fewer training samples, we can efficiently fine-tune LLMs and achieve competitive results.
> It is worth mentioning that our findings align with what reviewer 6ciM observed in their previous experiments in hyperparameter tuning for table fine-tuning, which demonstrates the applicability of our analysis in practice.
>
>
> Finally, as noted by reviewer 8ejo04, our TAMA model provides a valuable hyperparameter baseline for future table LLM research.
>
> We appreciate your thoughtful feedback and believe that your comments will encourage readers to engage critically with our work, ultimately benefiting our community.
> In the meantime, we start our paper with the motivation of emphasizing the technical details over telling a compelling story, as these details are crucial as revealed by our experiments and align with the observations in practice as pointed out by reviewers 8ejo04 and 6ciM. We hope our work will assist practitioners in building models more efficiently and effectively, ultimately benefiting the table research community and research for domain-specific instruction tuning in the longer run.
>
>
> ### References
>
> [1] Su, Aofeng, et al. "TableGPT2: A Large Multimodal Model with Tabular Data Integration." arXiv preprint arXiv:2411.02059 (2024).

---

> > ### Comment · Reviewer_nNeC · 2024-11-19
> >
> > Thanks for your rebuttal! I really appreciate your effort in explaining the contributions of your work.
> >
> > Unfortunately, after reading your rebuttal carefully, I remain unconvinced. In my understanding, Table LLM is just a very specialized downstream model, where its application scenarios exist but are relatively limited. Besides, it is also always expected that a quite specialized LLM's general capabilities might be hurt, so the result is not that surprising (at least to myself).
> >
> > The hyperparameter tuning you show is great. But overall, empirically evaluating a very specialized LLM and training a better LLM by just hyperparameter tuning seems to make the paper more like a good technical report, instead of a research paper that studies a scientific problem.
> >
> > I am still open to different opinions. Feel free to reply again if you want to convince me.

---

> > > ### Author Response · Authors · 2024-11-20
> > >
> > > We appreciate your engagement in the discussion! We would like to comment on your concerns.
> > >
> > > > TableLLM is a very specialized downstream model, where its application scenarios exist but are relatively limited.
> > >
> > > We respectfully disagree with the characterization that TableLLM's application scenarios are relatively limited. Unlike unstructured text, tables systematically organize large amounts of information, serving as the foundation for numerous applications, including medical diagnostics [3], virtual personal assistants, and customer relationship management systems [1, 2]. To handle various scenarios for table data, numerous table-related datasets have been proposed from the community, including table question answering datasets such as WikiTQA [4], table-to-text generation datasets such as ToTTo [5], table fact verification datasets such as TabFact [6]. To tackle these tasks, researchers have been putting a significant amount of effort into building strong table models [7, 8, 9, 10]. The diversity and the high impact of these table-related datasets prove the wide application scenarios for TableLLM.
> > >
> > > [1][The ATIS spoken language systems pilot corpus.](https://aclanthology.org/H90-1021.pdf)
> > >
> > > [2] [Expanding the scope of the ATIS task: The ATIS-3 corpus.](https://aclanthology.org/H94-1010.pdf)
> > >
> > > [3] [PubHealthTab: A public health table-based dataset for evidence-based fact checking.](https://aclanthology.org/2022.findings-naacl.1/)
> > >
> > > [4] [Compositional semantic parsing on semi-structured tables.](https://arxiv.org/abs/1508.00305)
> > >
> > > [5] [ToTTo: A controlled table-to-text generation dataset.](https://arxiv.org/abs/2004.14373)
> > >
> > > [6] [Tabfact: A large-scale dataset for table-based fact verification.](https://arxiv.org/abs/1909.02164)
> > >
> > > [7] [TaBERT: Pretraining for joint understanding of textual and tabular data](https://arxiv.org/pdf/2005.08314)
> > >
> > > [8] [TaPas: Weakly supervised table parsing via pre-training](https://arxiv.org/pdf/2004.02349)
> > >
> > > [9] [Tableformer: Robust transformer modeling for table-text encoding](https://arxiv.org/pdf/2203.00274)
> > >
> > > [10] [Unifiedskg: Unifying and multi-tasking structured knowledge grounding with text-to-text language models](https://arxiv.org/pdf/2201.05966)
> > >
> > > > It is always expected that a quite specialized LLM’s general capabilities might be hurt, so the result is not that surprising.
> > >
> > > While it is expected that specialization may compromise general capabilities, our work goes further by studying this trade-off **quantitatively**. We believe that our analysis provides a more holistic understanding of the limitations and potential of table LLMs.
> > >
> > > Moreover, contrary to the expectation that a highly specialized LLM's general capabilities will inevitably suffer, our exploration reveals that, with proper hyperparameter selection, it is possible to strike a balance between enhancing table understanding abilities and maintaining strong general capabilities.
> > >
> > > > The paper looks more like a good technical report, instead of a research paper that studies a scientific problem.
> > >
> > > We appreciate that you acknowledge the solidness of our paper.
> > >
> > > However, we respectfully disagree with the characterization of our paper as merely a technical report. Our research addresses a scientific problem: how to optimize table understanding while preserving general capabilities. And through our paper, we demonstrate that smaller learning rates and fewer training instances can achieve this balance, resulting in a model that performs well on table-specific tasks without sacrificing broader competencies.
> > >
> > > Furthermore, our systematic exploration of hyperparameter effects provides insights into the relationship between model behavior and training settings. This understanding contributes to the broader knowledge of how to design models that balance specialization and generalization effectively.

---

> > > > ### Comment · Reviewer_nNeC · 2024-11-20
> > > >
> > > > Thanks for your reply!
> > > >
> > > > Let me further clarify my judgment. Basically, I think tuning hyperparameters is a quite common (and almost always) practice when people try to train a new model. As you said, the solution to tackle the problem you studied (*how to optimize table understanding while preserving general capabilities*) is using this kind of common practice. Thus, I argue that the paper is more like a technical report (experiment report) as it mainly employs a common practice and reports the empirical findings.
> > > >
> > > > Overall I don't deny the contribution of your work to the open-source llm community (we always need people to show empirical experience about how to train better models), but the contribution of the paper itself looks relatively trivial so I am not convinced that it is a research paper for a ML conference.

---

> > > > > ### Author Response · Authors · 2024-11-20
> > > > >
> > > > > We appreciate your engagement in the discussion!
> > > > >
> > > > > First, we appreciate that you acknowledge the contribution of our work to the open-source LLM community.
> > > > >
> > > > > In terms of your concern,
> > > > >
> > > > > >  I think tuning hyperparameters is a quite common (and almost always) practice when people try to train a new model. As you said, the solution to tackle the problem you studied (how to optimize table understanding while preserving general capabilities) is using this kind of common practice.
> > > > >
> > > > > We would like to establish a common ground: the problem we study— *how to optimize table understanding while preserving general capabilities* —is indeed a valid and significant scientific research problem, and you are concerned that our solution to this problem is relatively trivial.
> > > > >
> > > > > However, this kind of systematic exploration has been largely overlooked in the existing literature. Moreover, existing literature uses significantly more training data in their fine-tuning stage for table-related tasks (e.g. as listed in Table 1, TableLlama has used up to 2 million datapoints). We are the first to show that proper training setups and fewer training instances can result in a model that performs well on table-specific tasks without sacrificing broader competencies.
> > > > >
> > > > >
> > > > > Last but not least, we would like to emphasize the three folds of contributions for our paper.
> > > > >
> > > > > - **Examining Overspecialization**: We provide a systematic examination of the overspecialization issues in existing table LLMs. We believe this is valuable for the community as it highlights the limitations and potential of current models.
> > > > >
> > > > > - **Systematic Study of Training Setups**: We investigate how different training configurations influence model behavior. Our findings show that with proper setups, we can maintain a model’s general capabilities while improving its abilities to deal with table tasks, offering practical guidelines for efficient model development.
> > > > >
> > > > > - **Open-Source Contribution**: As you noted, our model is a meaningful addition to the open-source LLM community, providing a strong baseline and facilitating future research in domain-specific instruction tuning.

---

> > > > > > ### Comment · Reviewer_nNeC · 2024-11-20
> > > > > >
> > > > > > Thanks for your reply! I also really appreciate your engagement and insights.
> > > > > >
> > > > > > I didn't deny that *this kind of systematic exploration has been largely overlooked in the existing literature*, and actually, I totally agree with this point. My main doubt is about whether this systematic exploration by just doing hyperparameter tuning (a common practice, which is more about engineering effort instead of novel techniques/theories/empirical methods/etc) could justify paper acceptance for an ML conference. Again, this effort seems to be more like a great contribution from a technical report, which could also fill the existing literature gap - filling the existing gap would not be a justification because IMO **how** to fill matters more.
> > > > > >
> > > > > > Thanks again for your engagement :-)

---

> ### Author Response · Authors · 2024-11-21
>
> We appreciate your continual engagement in this discussion. Though we respectfully disagree, we believe such conversation will help readers critically evaluate our contributions, ultimately benefiting the community in the longer run.
>
>
> First, we are glad that we establish the common ground that *our systematic exploration has been largely overlooked in the existing literature*. And we appreciate that you acknowledge *our paper can fill the existing literature gap*.
>
> Regarding your concern:
>
> > such a practice is more of an engineering effort, which you do not think can justify the acceptance for an ML conference.
>
> While we acknowledge that everyone has their own taste and perspective of research, we respectfully disagree with the characterization that contributions primarily grounded in engineering and empirical exploration are insufficient for ML conferences.
> Many impactful works published at ML venues advance our understanding through systematic empirical studies or engineering insights that fill important knowledge gaps.
> For example:
>
> - [1] published at Neurips explores how different data and models affect model capabilities after instruction tuning, offering valuable insights into instruction tuning.
>
> - [2] published at Neurips systematically evaluates popular architectures back then for table tasks, setting up baselines for future research.
>
> These works demonstrate the value of systematic empirical exploration in shaping the direction of follow-up research, and we are deeply inspired by such systematic exploration that these works have carried out.
>
> We would like to emphasize our contributions as follows:
>
> - **Examining Overspecialization**: We provide a systematic examination of the overspecialization issues in existing table LLMs. We believe this is valuable for the community as it highlights the limitations and potential of current models.
>
> - **Systematic Study of Training Setups**: We investigate how different training configurations influence model behavior. Our findings show that with proper setups, we can maintain a model’s general capabilities while improving its abilities to deal with table tasks, offering practical guidelines for efficient model development. As you noted, our systematic exploration has been largely overlooked in the existing literature, and our paper can fill the existing literature gap.
>
> - **Open-Source Contribution**: As you noted, our model is a meaningful addition to the open-source LLM community, providing a strong baseline and facilitating future research in domain-specific instruction tuning.
>
>
> ### References
>
> [1] [How far can camels go? exploring the state of instruction tuning on open resources.](https://proceedings.neurips.cc/paper_files/paper/2023/file/ec6413875e4ab08d7bc4d8e225263398-Paper-Datasets_and_Benchmarks.pdf)
>
> [2] [Revisiting deep learning models for tabular data.](https://proceedings.neurips.cc/paper_files/paper/2021/file/9d86d83f925f2149e9edb0ac3b49229c-Paper.pdf)

---

> > ### Comment · Reviewer_nNeC · 2024-11-24
> >
> > Thanks for your follow-up! Yeah, I agree that we may have different research tastes and opinions about what papers should/can be accepted to an ML conference, and I really appreciate your respect and patience.
> >
> > My final opinion is that this paper seems to be right on the threshold, and my score would be something like "5.5" (but it still leans towards 5 instead of 6 a little bit if I am "forced" to give a very clear preference). I would not support this paper's acceptance, but I would also not fight against its acceptance at all.
> >
> > Thanks again for your engagement! And again, I always appreciate your contribution to the open-source LLM community.

---

> > > ### Author Response · Authors · 2024-11-24
> > >
> > > We appreciate your continual engagement in the discussion, and we enjoy this intellectually stimulating process!
> > >
> > > We fully respect your final opinion, and sincerely thank you for expressing that *your final score would be something like 5.5*, and *you would not fight against its acceptance at all*. We understand the scoring system can be tricky.
> > >
> > > Through this exchange, we believe we have achieved many common grounds, with our contributions acknowledged by both you and other reviewers.
> > > We are glad to see our contributions are valued by our community.
> > > To summarize our contributions:
> > >
> > > - **Examining Overspecialization**: We provide a systematic examination of the overspecialization issues in existing table LLMs, which is valuable for the community as it highlights the limitations and potential of the existing models.
> > >
> > > - **Systematic Study of Training Setups**: We investigate how different training configurations influence model behavior. Our findings show that with proper setups, we can maintain a model’s general capabilities while improving its abilities to deal with table tasks, offering practical guidelines for efficient model development. Our systematic exploration has been largely overlooked in the existing literature, and our paper can fill the existing literature gap.
> > >
> > > - **Open-Source Contribution**: Our model is a meaningful addition to the open-source LLM community, providing a strong baseline and facilitating future research in domain-specific instruction tuning.

---

### Official Review · Reviewer_8ejo · 2024-11-04

**Soundness:** 3
**Presentation:** 3
**Contribution:** 2
**Rating:** 6
**Confidence:** 4

**Summary:**

1. This paper presents a comprehensive study revealing that current open-source table LLMs tend to be overly specialized for table-specific tasks, leading to a decrease in out-of-domain table understanding, instruction-following accuracy, and the general capabilities of the base model. These models, such as TableLLM and TableBenchLLM, typically employ high learning rates (e.g., 2e-5), extended training epochs (e.g. 6 epochs), and massive datasets (often exceeding 2 million instances) that lack diversity. This approach results in significant performance drops—up to 20% on general benchmarks like AI2ARC—compared to their base models.

2. The authors argue that careful hyperparameter tuning is necessary to balance table understanding with general capabilities. Through detailed experiments on tasks like FeTaQA, HiTab, and TabFact, they demonstrate how hyperparameters such as learning rate, epochs, and training sample size affect table LLM performance. As a result, they find that a learning rate of 1.0e-6 and two training epochs optimize the LLaMA 3.1-8B model for both table and general task performance.

3. Based on these findings, the authors sample 200 data instances each from 12 diverse table tasks, including WikiSQL, TATQA, and FEVEROUS, totaling only 2,600 samples—just 0.1% of the data typically used in other table LLMs. This data reduction, achieved by randomly sampling representative examples from each task, allows them to train TAMA, a new table LLM. TAMA matches or surpasses GPT-3.5 and, in some cases, GPT-4 on standard benchmarks, achieving 77.4% on FEVEROUS (3% higher than TableLLM) and 66.9% on MMLU, close to the base model’s score. TAMA’s efficient tuning strategy preserves broad capabilities, offering a resource-efficient yet powerful approach to table instruction tuning.

**Strengths:**

1. Identification of Over-Specialization Issues in Table LLMs: This work reveals a critical trend in current table LLMs, where excessive fine-tuning for table-specific tasks often compromises the models' generalization capabilities. By shedding light on the potential trade-offs, the study encourages the community to reconsider whether the performance gains claimed in state-of-the-art table LLMs justify the loss of general capabilities. This insight points to a promising direction for future research on balancing task-specific and general abilities in LLMs.

2. Extensive Experiments: The study conducts an extensive range of experiments across multiple tasks, including FeTaQA, HiTab, TabFact, and others, providing a broad evaluation of hyperparameter impacts on table-specific and general capabilities. This depth of experimentation allows for a more complete view of the challenges in table LLM tuning and establishes a robust foundation for future research in this area.

3. Identification of Effective Hyperparameter Settings: Through systematic exploration of hyperparameters—such as learning rate, training sample size, and number of epochs—the authors identify an optimal configuration for the LLaMA 3.1-8B model that supports strong table-specific performance while preserving general capabilities. By using significantly fewer training samples, they demonstrate an efficient tuning approach that achieves competitive results, establishing a valuable hyperparameter baseline for future table LLM research.

**Weaknesses:**

1. Limited Reproducibility Due to Closed Code and Weights: While the study’s findings are promising, the lack of open-source training and evaluation scripts, data, and TAMA’s model parameters during the review period restricts reproducibility. Although the study’s conclusions align with similar issues I observed in table LLMs like TableLLaMA and TableLLM, having access to TAMA’s scripts and parameters would allow a comprehensive verification of the training process and performance claims of TAMA, underscoring the importance of direct open-sourcing resources for rigorous validation.

2. Title-Content Discrepancy on Table Data Modality: The title, “Rethinking Table Instruction Tuning,” suggests a focus on unique aspects of table data, yet the study does not examine if the observed trade-offs are specific to table modality or generalizable. Discussion on the influence of table-specific features, such as heterogeneity, permutation invariance, or numeric density, could better align the study’s scope with the title.

3. Limited Novelty and Incomplete Analysis Relative to Zhou et al. (2024): While this work applies Zhou et al. (2024) [1]’s approach of using a small set of tuning data to enhance table LLM capabilities, as reflected in lines 246-249, it offers limited additional insights. Although the application in the table domain adds value, the study could have further clarified whether observed improvements are due to pre-trained knowledge or specific learning during fine-tuning.
Additionally, it does not explore whether models with stronger general capabilities require less tuning data to maximize performance in table tasks, which could provide valuable insights for future table LLM research. Finally, examining differences in pre-training corpora, such as LLaMA 1 vs. LLaMA 2 vs. LLaMA 3.1, could clarify whether observed gains stem from pre-trained abilities or are specific to instruction tuning. More experiments, such as comparing results across diverse pre-trained models, would enhance originality and provide practical insights.

4. Lack of Validation Across Multiple Base Models and Hyperparameter Guidance: While the hyperparameter settings for LLaMA 3.1-8B provide a useful baseline, extending these findings to other mainstream base models is essential to confirm generalizability. Applying the study’s tuning approach across diverse models, such as Mistral [2] and Qwen [3], and establishing a set of reliable hyperparameter baselines across mainstream LLMs—much like the impact achieved by the work [4]—would add significant, long-term value. Such efforts would enhance the study’s applicability and offer future table LLM research a practical framework for effective tuning.

[1] Zhou, Chunting, et al. "Lima: Less is more for alignment." Advances in Neural Information Processing Systems 36 (2024).

[2] https://huggingface.co/mistralai

[3] https://huggingface.co/Qwen

[4] Gorishniy, Yury, et al. "Revisiting deep learning models for tabular data." Advances in Neural Information Processing Systems 34 (2021): 18932-18943.

**Questions:**

I really appreciate the motivation and ideas presented in this work, and addressing the questions raised in the weaknesses section would support a higher rating. I am curious whether the conclusion that a small number of samples can achieve good results is based on fine-tuning for specific fields. Because the SFT in these subdivided tableqa fields seems to be the same as what we did in the BERT era, so large-scale pre-training + small-scale fine-tuning can achieve good results. I am curious whether this conclusion still holds true when large models need general capabilities such as reasoning, long text answers, and RAG?

I know it's not easy to answer the question. It would be appreciated if some insights can be provided.

---

> ### Author Response · Authors · 2024-11-19
>
> We appreciate your careful and constructive review.
> We thank you for pointing out that our work identifies the over-specialization issues in existing table LLMs, our experiments are comprehensive, and our research establishes a valuable hyperparameter baseline for future research.
>
> In terms of your concerns,
>
> > Closed Code and Weights
>
> We will release our script and model weights upon acceptance. Currently, we did not archive our paper or release any relevant resources online for the sake of integrity in the reviewing process. We want to avoid potential plagiarism or biasing reviewers after making our information public. We hope you can understand our reasoning.
>
> > does not examine if the observed trade-offs are specific to table modality or generalizable, such as heterogeneity, permutation invariance, or numeric density.
>
> We have expanded our analysis to assess how table-specific features may influence model performance.
> To investigate this, we leverage our results of training the Llama 3.1 8B Instruct model for three epochs using 500 examples on each dataset, respectively.
> We present our results as follows:
>
> |         | Number density in table (%) | Cells containing no number : Cells containing numbers | Total input tokens | Table tokens | Question tokens | Table tokens : Question tokens | MMLU  (lr = 1e-6) | MMLU (lr = 5e-6) | MMLU (lr = 1e-5) | IFEval (lr = 1e-6)  | IFEval (lr = 5e-6) | IFEval (lr = 1e-5) |
> |---------|-----------------------------|-------------------------------------------------------|--------------------|--------------|-----------------|--------------------------------|-------------------|------------------|------------------|---------------------|--------------------|--------------------|
> | TabFact | 73.03                       | 1.34 : 1                                              | 292,822            | 264,520      | 19,286          | 13.72 : 1                      | 66.74             | **64.51**            | **29.95**            | 77.70               | **49.40**              | **25.66**              |
> | FeTaQA  | 57.99                       | 1.68 : 1                                              | 309,624            | 251,697      | 42,492          | 5.92 : 1                       | 65.79             | 65.66            | 63.73            | 77.82               | 53.36              | 31.41              |
> | HiTab   | 80.60                       | 1.19 : 1                                              | 452,149            | 424,941      | 11,030          | 38.53 : 1                      | 66.37             | 66.77            | 62.91            | 78.18               | 49.40              | 29.74              |
>
> We find that the performance degradation is most significant on TabFact.
> Interestingly, despite TabFact having intermediate numeric density and table-to-question token ratios, it still shows the fastest performance decline.
>
> We hypothesize that this is due to the nature of the task rather than the table-specific features examined.
> Since FeTaQA and HiTab are table QA tasks, they may possess similar QA form that the model has encountered in its general instruction tuning stage, this may ease the decay of the model’s general capabilities in our fine-tuning stage.
> However, TabFact is about fact-checking, the input form includes both the table and the claim to be verified, which we suspect may not be as common as the QA data in its general instruction tuning stage.
> Therefore, the model suffers a more significant performance decay because it needs to update more of its internal knowledge to handle such a task.
>
> > further clarified whether observed improvements are due to pre-trained knowledge or specific learning during fine-tuning
>
> We believe that TAMA model has gained ability into handling table data in its fine-tuning stage based on the performance improvement on the out-of-domain test datasets (denoted as S1 and S2 in the paper).
> Specifically, we choose S1 and S2 as they are synthesized test sets by LLMs from existing literature [1, 2] compared to the training data where all the data is sourced from existing human-annotated benchmarks.
> The performance on these two out-of-domain datasets is:
>
> |        | **S1** | **S2** |
> |--------|-----------|------------|
> | **base** | 53.60     | 23.47     |
> | **TAMA** | **64.93** | **28.60**   |
>
> While pre-training certainly imparts a foundational understanding of table-related knowledge, our results indicate that table-specific fine-tuning plays a crucial role in further enhancing the model’s capability in handling table data.
>
> [1] Li, Peng, et al. "Table-gpt: Table-tuned gpt for diverse table tasks." arXiv preprint arXiv:2310.09263 (2023).
>
> [2] Wu, Xianjie, et al. "TableBench: A Comprehensive and Complex Benchmark for Table Question Answering." arXiv preprint arXiv:2408.09174 (2024).

---

> > ### Author Response · Authors · 2024-11-19
> >
> > > whether models with stronger general capabilities require less tuning data
> >
> > We conduct additional experiments with respect to Llama 2 7B Instruct model, training on the 1,500 examples from HiTab, FeTaQA, and TabFact.
> > The table below presents the results at a learning rate of 1.0e-6, which we select based on our answer to your next question for Llama 2 7B Instruct:
> >
> > | Training Data   |   FeTaQA |   TabFact |   MMLU |   IFEval |
> > |:----------------:|---------:|----------:|-------:|---------:|
> > | 30 |    13.32 |     31.68|  47.07 |    45.08 |
> > | 90 |    13.86 |     49.51 |  46.96 |    46.16 |
> > | 150 |    14.79 |     46.24 |  47.09 |    47.48 |
> > | 300 |    14.47 |     50.27 |  47.09 |    45.56 |
> > | 600 |    24.12 |     50.74 |  47.11 |    45.56  |
> > | 1500 |    29.03 |     53.80 |  47.07 |    47.84 |
> >
> > We find that given the same number of training instances, Llama 3.1 8B Instruct achieves better performance than Llama 2 7B Instruct.
> > For instance, when trained with the same 1,500 examples at the learning rate of 1.0e-6, Llama 3.1 8B Instruct yields 73.10 on TabFact while Llama 2 7B Instruct only yields 53.80.
> >
> > Therefore, models with stronger general capabilities require less tuning data.
> >
> > > Validation across models.
> >
> > We have conducted additional experiments to validate our findings across different models in the full-parameter setup, including Llama 2 7B Instruct, QWen 2.5 7B Instruct, Mistral v0.3 7B Instruct, and Phi 3 small 8K Instruct (7B).
> >
> >
> > ### **Learning Rate Analysis**
> >
> > We train each model on 500 examples from HiTab, FeTaQA, and TabFact (1,500 examples total) for three epochs. Below are the results for different learning rates:
> >
> > Llama 2 7B Instruct:
> >
> > | Learning Rate | FeTaQA     | TabFact     | MMLU | IFEval  |
> > |---------------|-------------|-------------|---------|-------------|
> > | 5.00E-07      | 26.54 | 52.63 | 47.12   | 47.84 |
> > | 1.00E-06      | 29.03 | 53.80 | 47.07   | **47.84** |
> > | 5.00E-06      | 33.86 | 51.05 | 46.58   | **35.25** |
> > | 1.00E-05      | 34.77 | 53.79 | 45.99   | 39.93 |
> >
> >
> > QWen 2.5 7B Instruct:
> >
> > | Learning Rate | FeTaQA     | TabFact      | MMLU | IFEval  |
> > |---------------|-------------|-------------|---------|-------------|
> > | 5.00E-07      | 33.14   | 71.09   | 73.66  | 76.02   |
> > | 1.00E-06      | 34.50   | 72.66   | 73.52   | **75.78**   |
> > | 5.00E-06      | 34.04   | 72.81   | 73.81   | **49.28**   |
> > | 1.00E-05      | 33.84   | 71.51   | 73.49   | 41.61   |
> >
> >
> >
> > Mistral v0.3 7B Instruct:
> >
> > | Learning Rate | FeTaQA     | TabFact    | MMLU | IFEval  |
> > |---------------|-------------|-------------|---------|-------------|
> > | 1.00E-07      | 31.91   | 64.32   | 61.32   | 62.83   |
> > | 5.00E-07      | 36.44   | 70.35   | 60.76   | 57.79   |
> > | 1.00E-06      | 36.99   | 71.88   | 60.45   | **52.28**   |
> > | 5.00E-06      | 35.71   | 53.64   | 34.96   | **33.09**   |
> > | 1.00E-05      | 32.14   | 50.87   | 24.93   | 27.70   |
> >
> >
> >
> > Phi 3 8K Instruct (7B):
> >
> > | Learning Rate | FeTaQA     | TabFact        | MMLU | IFEval  |
> > |---------------|-------------|-------------|---------|-------------|
> > | 1.00E-06      | 33.10   | 72.04   | 70.48   | 71.22   |
> > | 5.00E-06      | 37.26   | 73.82   | 74.89   | 68.71   |
> > | 1.00E-05      | 38.13   | 73.92   | 73.30   | **62.95**   |
> > | 5.00E-05      | 34.46   | 50.90   | 49.08   | **28.78**   |
> > | 1.00E-04      | 30.66   | 50.33   | 49.17   | 23.02   |
> >
> >
> >
> > Key Findings:
> >
> > **Performance Drop**: We observe a significant performance drop happens for every model on the two general benchmarks. Interestingly, for models such as QWen 2.5, when we increase the learning rate from 1.0e-6 to 5.0e-6, it would primarily affect the IFEval dataset rather than MMLU, suggesting that the compromises may happen at different speeds with respect to different aspects of the model's general capability.
> >
> > **Different “Breakdown Point”**: The Phi model shows a pronounced performance drop from 1.0e-5 to 5.0e-5, in contrast to Llama, Mistral and QWen models, where the “breakdown point” on the learning rate is slightly smaller, especially for Mistral model, where we see 5 points lose on IFEval from 5.0e-7 to 1.0e-6.
> > Here we list the learning rate we would suggest for practitioners to use if they would fine-tune the following LLMs on table-specific tasks:
> >
> > | Model                          | Learning Rate     |
> > |--------------------------------|-------------------|
> > | Llama 2 7B Instruct            | 1.0e-6 / 5.0e-7   |
> > | Llama 3.1 8B Instruct          | 1.0e-6 / 5.0e-7   |
> > | QWen 2.5 7B Instruct           | 1.0e-6 / 5.0e-7   |
> > | Mistral v0.3 7B Instruct       | 5.0e-7 / 1.0e-7   |
> > | Phi 3 small 8K Instruct (7B)   | 5.0e-6 / 1.0e-6   |

---

> ### Author Response · Authors · 2024-11-19
>
> ### **Training Size Analysis**
>
> We further experiment with various training sizes for each model to observe the impact on performance. Below are the results for Llama 2, QWen 2.5, Mistral v0.3, and Phi 3 8K models at one of the learning rates we select based on our results in the Learning Rate Analysis when we train them for three epochs.
>
>
> Llama 2 7B Instruct (Learning Rate: 1.0e-6):
>
>
> | Training Data   |   FeTaQA |   TabFact |   MMLU |   IFEval |
> |:----------------:|---------:|----------:|-------:|---------:|
> | 30 |    13.32 |     31.68|  47.07 |    45.08 |
> | 90 |    13.86 |     49.51 |  46.96 |    46.16 |
> | 150 |    14.79 |     46.24 |  47.09 |    47.48 |
> | 300 |    14.47 |     50.27 |  47.09 |    45.56 |
> | 600 |    24.12 |     50.74 |  47.11 |    45.56  |
> | 1500 |    29.03 |     53.80 |  47.07 |    47.84 |
>
> QWen 2.5 7B (Learning Rate: 1.0e-6):
>
> | Training Data   |   FeTaQA |   TabFact|   MMLU |   IFEval |
> |:----------------:|---------:|----------:|-------:|---------:|
> |           30   |    14.2  |      8.42 |  73.91 |    70.43 |
> |           90   |    16.45 |      8.47 |  73.76 |    70.43 |
> |           150   |    21.14 |     69.66 |  73.83 |    69.5  |
> |           300  |    22.1  |     69.65 |  73.72 |    68.95 |
> |           600  |    32.12 |     70.86 |  73.71 |    68.21 |
> |           1500  |    34.5  |     72.66 |  73.52 |    66.73 |
>
>
> Mistral v0.3 7B Instruct (Learning Rate: 5.0e-7):
>
> | Training Data   |   FeTaQA |   TabFact |   MMLU |   IFEval |
> |:----------------|---------:|----------:|-------:|---------:|
> | 30   |    23.84 |      0.28 |  61.39 |    49.72 |
> | 90   |    10.67 |     60.29 |  61.34 |    51.76 |
> | 150   |    19.79 |     49.82 |  61.34 |    52.87 |
> | 300  |    33.93 |     61.91 |  61.13 |    51.02 |
> | 600  |    34.28 |     66.34 |  61.12 |    52.31 |
> | 1500  |    36.44 |     70.35 |  60.76 |    47.69 |
>
> Phi 3 8K Instruct (7B) (Learning Rate: 5.0e-6):
>
> | Training Data   |   FeTaQA |   TabFact |   MMLU |   IFEval |
> |:----------------|---------:|----------:|-------:|---------:|
> | 30   |    17.19 |      9.62 |  75.43 |    52.31 |
> | 90   |    24.01 |     67.32 |  75.43 |    63.96 |
> | 150   |    24.67 |     68    |  75.43 |    62.11 |
> | 300  |    34.81 |     71.3  |  75.61 |    62.85 |
> | 600  |    37.74 |     72.91 |  75.5  |    61.18 |
> | 1500  |    37.26 |     73.82 |  75.26 |    59.7  |
>
> Key Findings:
>
> Across all models, performance improvement becomes marginal from 600 to 1500 examples, suggesting diminishing returns with larger datasets.
>
>
> > Whether this conclusion still holds true when large models need general capabilities such as reasoning, long text answers, and RAG?
>
> We appreciate your question and would like to share our thoughts here.
>
> In our ongoing works, we have experimented with using a small sample of randomly selected training instances on an NLI explanation dataset. We have observed similar trends: even with only around 10% of the original dataset, Llama 3.1 8B Instruct achieved competitive scores on the test set.
>
> We attribute the phenomenon of training LLMs on limited data still leads to competitive performance to two main factors.
> First, today’s LLMs are more capable than ever – they can easily pick up patterns, sometimes at the cost of overfitting rather than generalizing broadly. In addition, these models already possess a substantial amount of knowledge and abilities from extensive pre-training, which gives them a head start in fine-tuning.
>
> Second, many tasks share similar reasoning paths. For instance, a math QA dataset may predominantly feature questions about averaging, counting, or addition, and a strong base model may master them with a few examples. This leads to their competitive performance even with limited data.
>
> With these reasons above, we think that for tasks requiring more diverse and complex reasoning paths, or combination of distinct abilities, more training examples may be helpful. The challenge lies in the fact that it is hard to construct a comprehensive dataset – domain-specific tasks often lack diversity, and general benchmarks like RAG or long-text generation require substantial manual labor to craft a comprehensive, diverse and high-quality dataset. Designing high-quality reasoning benchmarks is also difficult, as it requires expert annotators who can think creatively and cover a wide range of reasoning skills.
>
>
> We have also updated our manuscript accordingly, where the content in blue corresponds to the content we add.

---

> > ### Comment · Reviewer_8ejo · 2024-11-25
> > **Thanks for your responses**
> >
> > I greatly appreciate the authors' thorough experiments and detailed rebuttal.
> >
> > We find that the performance degradation is most significant on TabFact. Interestingly, despite TabFact having intermediate numeric density and table-to-question token ratios, it still shows the fastest performance decline.
> > We hypothesize that this is due to the nature of the task rather than the table-specific features examined.
> > 1. Regarding the statement: “We hypothesize that this is due to the nature of the task rather than the table-specific features examined,” if no table-specific factors are identified as contributing to the observed trade-offs, it suggests that these trade-offs may be generalizable rather than specific to the table modality. Therefore, I maintain my concern about the Title-Content Discrepancy on Table Data Modality. The title, “Rethinking Table Instruction Tuning,” implies a focus on unique aspects of table data. I recommend revising the title to better reflect the study's broader scope and avoid potentially misleading professionals in the field.
> >
> > We will release our script and model weights upon acceptance. Currently, we did not archive our paper or release any relevant resources online for the sake of integrity in the reviewing process. We want to avoid potential plagiarism or biasing reviewers after making our information public. We hope you can understand our reasoning.
> > 2. I understand the authors' concerns. However, many anonymous code-sharing platforms, such as anonymous.4open.science, are frequently used during the ICLR review process. Uploading code to such platforms after removing personal information would maintain integrity in the review process while allowing reviewers to verify the results.
> >
> > Lastly, I have mixed thoughts on this paper. While it has empirical merits, the technical contribution of hyperparameter tuning is indeed not negligible. Moreover, some empirical findings have also been widely observed in LLM researcher's daily experiments. So I agree with Reviewer nNeC that this is a really borderline paper, I would not be upset if this article was accepted or rejected.

---

> ### Author Response · Authors · 2024-11-27
>
> We appreciate your engagement in the discussion and your suggestions, which we believe have made our experiments more comprehensive.
>
> **Regarding the paper title**
>
> We are currently discussing the title among the authors. One possible revision we are considering is “Rethinking Instruction Tuning: A Case Study on Table Data.” We believe this title better reflect the study's broader scope while emphasizing our experiments on table data. We aim to finalize the title after further internal discussions.
>
> **Regarding open-sourcing the script and model weights.**
>
> Thank you for pointing us to anonymous.4open.science! Currently, we are categorizing our script, dealing with the license issues for the data, and removing the identity information in our script.
>
> We understand and fully respect your attitude towards our paper and greatly appreciate your engagement in the discussion. Thank you for your constructive feedback, and we hope you have a wonderful Thanksgiving break if you celebrate it :)

---

### Official Review · Reviewer_6ciM · 2024-11-04

**Soundness:** 3
**Presentation:** 3
**Contribution:** 3
**Rating:** 8
**Confidence:** 4

**Summary:**

This paper presents a thorough evaluation of existing instruction-tuned LLM for table-related tasks, focusing on the often-overlooked aspects of hyper-parameter choices and their impact on model performance. The authors identify the clear declines in the general knowledge obtained by the base model. Through systematic analysis, this work demonstrates that proper learning rates and fewer training instances can enhance table understanding without the sacrifice of LLM's exiting capabilities. Based on these findings, the proposed TAMA can be trained with fewer examples and achieves impressive performance on several representative tabular tasks.

**Strengths:**

- Comprehensive Analysis: The paper provides a thorough analysis of existing table LLMs, highlighting the importance of hyperparameter tuning, which is often neglected in other studies.

- Empirical Findings: The findings are helpful to subsequential research in this field of applying LLMs into tabular tasks. The effects of learning rate is close to my previous experimence in hyper-parameter tuning while further training LLMs over tables.

- Based on the finding, the proposed TAMA demonstrates an impressive improvement in performance.

**Weaknesses:**

These experiments were only based on Llama models. Further examination is needed for other LLMs, e.g., Qwen-* series, Phi-* series, etc. Whether the findings still hold requires further analysis.

**Questions:**

Whether the findings still hold while applying lora or qlora?

---

> ### Author Response · Authors · 2024-11-19
>
> We thank you for pointing out that our analysis is comprehensive, our findings are helpful to subsequent research, and TAMA demonstrates an impressive improvement in performance.
>
> In terms of your concerns,
>
> > Experiments on other LLMs
>
> We have conducted additional experiments to validate our findings across different models in the full-parameter setup, including Llama 2 7B Instruct, QWen 2.5 7B Instruct, Mistral v0.3 7B Instruct, and Phi 3 small 8K Instruct (7B).
>
>
> ### **Learning Rate Analysis**
>
> We train each model on 500 examples from HiTab, FeTaQA, and TabFact (1,500 examples total) for three epochs. Below are the results for different learning rates:
>
> Llama 2 7B Instruct:
>
> | Learning Rate | FeTaQA     | TabFact     | MMLU | IFEval  |
> |---------------|-------------|-------------|---------|-------------|
> | 5.00E-07      | 26.54 | 52.63 | 47.12   | 47.84 |
> | 1.00E-06      | 29.03 | 53.80 | 47.07   | **47.84** |
> | 5.00E-06      | 33.86 | 51.05 | 46.58   | **35.25** |
> | 1.00E-05      | 34.77 | 53.79 | 45.99   | 39.93 |
>
>
> QWen 2.5 7B Instruct:
>
> | Learning Rate | FeTaQA     | TabFact      | MMLU | IFEval  |
> |---------------|-------------|-------------|---------|-------------|
> | 5.00E-07      | 33.14   | 71.09   | 73.66  | 76.02   |
> | 1.00E-06      | 34.50   | 72.66   | 73.52   | **75.78**   |
> | 5.00E-06      | 34.04   | 72.81   | 73.81   | **49.28**   |
> | 1.00E-05      | 33.84   | 71.51   | 73.49   | 41.61   |
>
>
>
> Mistral v0.3 7B Instruct:
>
> | Learning Rate | FeTaQA     | TabFact    | MMLU | IFEval  |
> |---------------|-------------|-------------|---------|-------------|
> | 1.00E-07      | 31.91   | 64.32   | 61.32   | 62.83   |
> | 5.00E-07      | 36.44   | 70.35   | 60.76   | 57.79   |
> | 1.00E-06      | 36.99   | 71.88   | 60.45   | **52.28**   |
> | 5.00E-06      | 35.71   | 53.64   | 34.96   | **33.09**   |
> | 1.00E-05      | 32.14   | 50.87   | 24.93   | 27.70   |
>
>
>
> Phi 3 8K Instruct (7B):
>
> | Learning Rate | FeTaQA     | TabFact        | MMLU | IFEval  |
> |---------------|-------------|-------------|---------|-------------|
> | 1.00E-06      | 33.10   | 72.04   | 70.48   | 71.22   |
> | 5.00E-06      | 37.26   | 73.82   | 74.89   | 68.71   |
> | 1.00E-05      | 38.13   | 73.92   | 73.30   | **62.95**   |
> | 5.00E-05      | 34.46   | 50.90   | 49.08   | **28.78**   |
> | 1.00E-04      | 30.66   | 50.33   | 49.17   | 23.02   |
>
>
>
> Key Findings:
>
> **Performance Drop**: We observe a significant performance drop happens for every model on the two general benchmarks. Interestingly, for models such as QWen 2.5, when we increase the learning rate from 1.0e-6 to 5.0e-6, it would primarily affect the IFEval dataset rather than MMLU, suggesting that the compromises may happen at different speeds with respect to different aspects of the model's general capability.
>
> **Different “Breakdown Point”**: The Phi model shows a pronounced performance drop from 1.0e-5 to 5.0e-5, in contrast to Llama, Mistral and QWen models, where the “breakdown point” on the learning rate is slightly smaller, especially for Mistral model, where we see 5 points lose on IFEval from 5.0e-7 to 1.0e-6.
> Here we list the learning rate we would suggest for practitioners to use if they would fine-tune the following LLMs on table-specific tasks:
>
> | Model                          | Learning Rate     |
> |--------------------------------|-------------------|
> | Llama 2 7B Instruct            | 1.0e-6 / 5.0e-7   |
> | Llama 3.1 8B Instruct          | 1.0e-6 / 5.0e-7   |
> | QWen 2.5 7B Instruct           | 1.0e-6 / 5.0e-7   |
> | Mistral v0.3 7B Instruct       | 5.0e-7 / 1.0e-7   |
> | Phi 3 small 8K Instruct (7B)   | 5.0e-6 / 1.0e-6   |

---

> ### Author Response · Authors · 2024-11-19
>
> ### **Training Size Analysis**
>
> We further experiment with various training sizes for each model to observe the impact on performance. Below are the results for Llama 2, QWen 2.5, Mistral v0.3, and Phi 3 8K models at one of the learning rates we select based on our results in the Learning Rate Analysis when we train them for three epochs.
>
>
> Llama 2 7B Instruct (Learning Rate: 1.0e-6):
>
>
> | Training Data   |   FeTaQA |   TabFact |   MMLU |   IFEval |
> |:----------------:|---------:|----------:|-------:|---------:|
> | 30 |    13.32 |     31.68|  47.07 |    45.08 |
> | 90 |    13.86 |     49.51 |  46.96 |    46.16 |
> | 150 |    14.79 |     46.24 |  47.09 |    47.48 |
> | 300 |    14.47 |     50.27 |  47.09 |    45.56 |
> | 600 |    24.12 |     50.74 |  47.11 |    45.56  |
> | 1500 |    29.03 |     53.80 |  47.07 |    47.84 |
>
> QWen 2.5 7B (Learning Rate: 1.0e-6):
>
> | Training Data   |   FeTaQA |   TabFact|   MMLU |   IFEval |
> |:----------------:|---------:|----------:|-------:|---------:|
> |           30   |    14.2  |      8.42 |  73.91 |    70.43 |
> |           90   |    16.45 |      8.47 |  73.76 |    70.43 |
> |           150   |    21.14 |     69.66 |  73.83 |    69.5  |
> |           300  |    22.1  |     69.65 |  73.72 |    68.95 |
> |           600  |    32.12 |     70.86 |  73.71 |    68.21 |
> |           1500  |    34.5  |     72.66 |  73.52 |    66.73 |
>
>
> Mistral v0.3 7B Instruct (Learning Rate: 5.0e-7):
>
> | Training Data   |   FeTaQA |   TabFact |   MMLU |   IFEval |
> |:----------------|---------:|----------:|-------:|---------:|
> | 30   |    23.84 |      0.28 |  61.39 |    49.72 |
> | 90   |    10.67 |     60.29 |  61.34 |    51.76 |
> | 150   |    19.79 |     49.82 |  61.34 |    52.87 |
> | 300  |    33.93 |     61.91 |  61.13 |    51.02 |
> | 600  |    34.28 |     66.34 |  61.12 |    52.31 |
> | 1500  |    36.44 |     70.35 |  60.76 |    47.69 |
>
> Phi 3 8K Instruct (7B) (Learning Rate: 5.0e-6):
>
> | Training Data   |   FeTaQA |   TabFact |   MMLU |   IFEval |
> |:----------------|---------:|----------:|-------:|---------:|
> | 30   |    17.19 |      9.62 |  75.43 |    52.31 |
> | 90   |    24.01 |     67.32 |  75.43 |    63.96 |
> | 150   |    24.67 |     68    |  75.43 |    62.11 |
> | 300  |    34.81 |     71.3  |  75.61 |    62.85 |
> | 600  |    37.74 |     72.91 |  75.5  |    61.18 |
> | 1500  |    37.26 |     73.82 |  75.26 |    59.7  |
>
> Key Findings:
>
> Across all models, performance improvement becomes marginal from 600 to 1500 examples, suggesting diminishing returns with larger datasets.

---

> > ### Author Response · Authors · 2024-11-19
> >
> > > Whether the findings still hold for LoRA and QLoRA
> >
> > Yes, our findings still hold for LoRA and QLoRA.
> >
> > We have conducted additional experiments using LoRA and QLoRA based on Llama 3.1-8B-Instruct. Specifically, we use hugging-quants/Meta-Llama-3.1-8B-Instruct-AWQ-INT4 as the base model for our QLoRA experiments.
> >
> > We replicate the experiments from our response to your question before, and here we also present our results in two aspects, the learning rate and the number of examples.
> >
> > ### **Learning Rate Analysis**
> >
> > We train the model using 500 examples from HiTab, FeTaQA, and TabFact (1,500 examples in total). Below are the results:
> >
> >
> > LoRA Performance:
> >
> >
> > | Learning Rate | FeTaQA      | TabFact     | MMLU    | IFEval      |
> > |---------------|-------------|-------------|---------|-------------|
> > | 1.00E-06      | 16.63 | 63.21 | 66.06   | 80.22 |
> > | 5.00E-06      | 23.69 | 66.80 | 65.97   | 80.94  |
> > | 1.00E-05      | 29.66 | 68.58 | 66.03   | **80.58** |
> > | 5.00E-05      | 35.33 | 73.80 | 67.04   | **76.98** |
> > | 1.00E-04      | 35.81 | 75.63 | 67.42   | **71.22** |
> > | 5.00E-04      | 36.04 | 73.88 | 66.36   | **60.67** |
> > | 1.00E-03      | 35.54 | 73.64 | 59.02   | 38.73 |
> >
> >
> > QLoRA Performance:
> >
> >
> > | Learning Rate | FeTaQA | TabFact | MMLU | IFEval |
> > |---------------|---------|---------|---------|---------|
> > | 1.00E-07      | 20.36   | 63.06   | 64.56   | 80.22   |
> > | 5.00E-07      | 19.07   | 66.42   | 64.68   | 80.46   |
> > | 1.00E-06      | 27.44   | 67.18   | 64.68   | 79.98   |
> > | 5.00E-06      | 34.64   | 70.98   | 64.76   | 78.66   |
> > | 1.00E-05      | 36.86   | 73.20   | 65.22   | 77.58   |
> > | 5.00E-05      | 36.52   | 74.11   | 65.82   | 76.02   |
> > | 1.00E-04      | 35.94   | 74.91   | 65.76   | **74.22**   |
> > | 5.00E-04      | 33.72   | 50.50   | 42.76   | **32.85**   |
> > | 1.00E-03      | 0.01    | 50.16   | 22.95   | 23.86   |
> >
> >
> > Key Findings:
> >
> > We find that there is still a “breakdown point” where further increasing the learning rate causes a sharp decline in overall performance for both LoRA and QLoRA. However, such “breakdown point” for LoRA and QLoRA (around 5.0e-5) is larger than the full parameter tuning (usually around 1.0e-6).
> > When the learning rate does not surpass such a “breakdown point”, both methods demonstrate competitive in-domain performance on table tasks.

---

> > > ### Author Response · Authors · 2024-11-19
> > >
> > > ### **Training Size Analysis**
> > >
> > > We have also evaluated the impact of the number of training examples on performance. The following results correspond to training LoRA and QLoRA at one of the learning rates we select based on our results in ### Learning Rate Analysis when we train them for three epochs.
> > >
> > >
> > > LoRA Performance (Learning Rate: 5e-5):
> > >
> > >
> > > | Training Data  | FeTaQA | TabFact | MMLU  | IFEval |
> > > |----------------|--------|---------|-------|--------|
> > > | 30  | 17.36  | 63.89   | 66.14 | 71.90  |
> > > | 90  | 19.83  | 66.50   | 66.03 | 70.98  |
> > > | 150  | 14.69  | 68.62   | 66.10 | 73.01  |
> > > | 300 | 26.01  | 67.96   | 66.20 | 72.09  |
> > > | 600 | 34.08  | 72.13   | 66.65 | 70.61  |
> > > | 1500 | 35.33  | 73.80   | 67.04 | 68.39  |
> > >
> > >
> > > QLoRA Performance (Learning Rate: 5e-5):
> > >
> > >
> > > | Training Data  | FeTaQA | TabFact | MMLU  | IFEval |
> > > |------------------|--------|---------|--------|--------|
> > > | 30    | 18.02  | 66.55   | 64.78  | 72.46  |
> > > | 90    | 35.33  | 68.44   | 65.08  | 69.32  |
> > > | 150    | 33.50  | 69.78   | 65.36  | 74.31  |
> > > | 300   | 35.95  | 69.46   | 65.63  | 71.72  |
> > > | 600   | 36.25  | 73.68   | 65.80  | 69.13  |
> > > | 1500   | 36.52  | 74.11   | 65.82  | 65.62  |
> > >
> > >
> > >
> > > Key Findings:
> > >
> > > Similar to what we have found for full parameter fine-tuning, both LoRA and QLoRA show diminishing returns as the number of training examples increases. While performance improves with more examples, the rate of improvement slows beyond 600 examples for LoRA. For QLoRA, the rate of improvement slows beyond 90 examples.
> > >
> > > We find that with 1,500 examples, QLoRA and LoRA perform similarly on the in-domain table tasks, and on FeTaQA, QLoRA even outperforms LoRA by 1 point. This suggests that practitioners may leverage such parameter-efficient fine-tuning methods like QLoRA in practice, especially when they have limited table data.
> > >
> > > We have also updated our manuscript accordingly, where the content in blue corresponds to the content we add.

---

> > > > ### Comment · Reviewer_6ciM · 2024-11-25
> > > >
> > > > Thanks a lot for you response, solved my concerns. I will increase the rating.

---

> > > > > ### Author Response · Authors · 2024-11-25
> > > > >
> > > > > Thank you for your positive feedback! We are delighted that our response addressed your concerns.

---

### Author Response · Authors · 2024-11-27

As the discussion period is drawing to an end, we would like to sincerely thank all the reviewers again. We greatly appreciate your constructive feedback and the thoughtful discussions throughout this process. Your suggestions have improved the quality of our paper, and we are deeply grateful for your time and effort.
We wish you a wonderful Thanksgiving break if you celebrate it!

---

### Author Response · Authors · 2024-12-04

We sincerely thank all the reviewers for their engagement and thoughtful comments throughout the discussion period. We are deeply grateful for the reviewers’ time and effort.

We greatly appreciate the reviewer’s recognition of our contributions in

- A thorough analysis of existing table LLMs (noted by reviewers 6ciM, 8ejo, and nNeC).

- Extensive experiments across multiple tasks, providing a broad evaluation of hyperparameter impacts on table-specific and general capabilities (noted by reviewer 8ejo), which are valuable to subsequent research (noted by reviewer 6ciM and 8ejo).

- Demonstration of an efficient tuning approach that achieves competitive results (noted by reviewer 6ciM, 8ejo, and nNeC), which leads to our TAMA model.

In response to the reviewers’ suggestions, we have expanded our manuscript as follows:

- Examined the applicability of our findings to other LLMs (Appendix C.2) and training setups (Appendix C.3).

- Conducted a trade-off analysis of data properties (Appendix C.5).

Thank you again to everyone!

---

### Meta-Review · Area_Chair_ySTr · 2024-12-22

**Metareview:**

The paper evaluates existing table-based LLMs and finds they exhibit poor OOD table understanding, weak instruction following, and limited general capabilities. It then investigates the influence of hyperparameters on training such models. Finally, the paper introduces TAMA, trained from LLaMA-3.1-8B-Instruct, which demonstrates significantly improved table understanding while maintaining general capabilities.

The paper thoroughly analyzes existing table LLMs, emphasizing the importance of hyperparameter tuning, an aspect often overlooked in other studies. The findings are valuable for subsequent research on applying LLMs to tabular tasks. The observed effects of learning rate align with my previous experience in hyperparameter tuning for further training LLMs on tabular data. Based on these findings, the proposed TAMA demonstrates impressive performance improvements. TAMA is a valuable contribution to the open-source LLM community, offering strong performance and providing a foundation for future work exploring LLM capabilities in this domain.

**Additional Comments On Reviewer Discussion:**

After careful consideration of all reviews and rebuttals, I share reviewer nNeC's concern that this work resembles a strong technical report, albeit one that addresses a gap in the existing literature. Hyperparameter tuning is common practice when adapting a general LLM for a specific task, therefore more insightful findings regarding generalization would significantly strengthen the submission.

---

### Decision · Program_Chairs · 2025-01-22

Reject